# Telomeric 8-oxo-guanine drives rapid premature senescence in the absence of telomere shortening

Ryan P. Barnes[1,2], Mariarosaria de Rosa [1,2,6], Sanjana A. Thosar[1,2,6], Ariana C. Detwiler [1,2], Vera Roginskaya[2,3], Bennett Van Houten [2,3], Marcel P. Bruchez[4], Jacob Stewart-Ornstein[2,5] and Patricia L. Opresko [1,2,3] ✉

**Oxidative stress is a primary cause of cellular senescence and contributes to the etiology of numerous human diseases. Oxidative damage to telomeric DNA has been proposed to cause premature senescence by accelerating telomere shortening. Here, we tested this model directly using a precision chemoptogenetic tool to produce the common lesion 8-oxo-guanine (8oxoG) exclusively at telomeres in human fibroblasts and epithelial cells. A single induction of telomeric 8oxoG is sufficient to trigger multiple hallmarks of p53-dependent senescence. Telomeric 8oxoG activates ATM and ATR signaling, and enriches for markers of telomere dysfunction in replicating, but not quiescent cells. Acute 8oxoG production fails to shorten telomeres, but rather generates fragile sites and mitotic DNA synthesis at telomeres, indicative of impaired replication. Based on our results, we propose that oxidative stress promotes rapid senescence by producing oxidative base lesions that drive replication-dependent telomere fragility and dysfunction in the absence of shortening and shelterin loss.**

Mammalian telomeres consist of 5′-TTAGGG-3′ arrays bound by shelterin—a protein complex that remodels the chromosome end to suppress inappropriate recognition by DNA damage response (DDR) signaling[1]. Progressive telomere shortening with cell division activates the DDR and triggers 'replicative senescence' characterized by cell cycle arrest and phenotypic changes[2,3]. Thus, telomeres act as potent tumor suppressors by limiting proliferation[4]. However, senescent cells accumulate with age and contribute to numerous ageing-related pathologies by compromising regenerative capacity and secreting inflammatory cytokines, chemokines and proteases that promote inflammation and alter the tissue microenvironment[5]. The microenvironment becomes more permissive for tumor growth and, thus, paradoxically senescence can also promote tumorigenesis, metastasis or immunosuppression[6–8]. Telomere dysfunction in premalignant cells with compromised DDR signaling can cause chromosomal fusions and instability, which drive carcinogenesis[9,10]. Thus, telomere function and integrity are critical for genome stability, cellular function and organism health.

Numerous studies from human tissues, mice and cell culture show that chronic inflammation and oxidative stress associate with accelerated telomere shortening and dysfunction[11,12]. Oxidative stress, which occurs when reactive oxygen species (ROS) exceed antioxidants, can promote senescence, degenerative diseases and aging[13–15]. Guanine is the base most susceptible to oxidation, and TTAGGG repeats are preferred sites for production of the common oxidative lesion 8oxoG[16,17]. These data led to a model, proposed around 20 years ago, that oxidative modification to telomeric bases may contribute more to telomere loss and telomere-driven senescence than the end-replication problem[18]. ROS-induced damage was also proposed to explain telomere dysfunction arising in low-proliferative tissues, such as lung and heart, independently of telomere length changes[19–23]. Infiltrating neutrophils in liver trigger senescence in neighboring hepatocytes via ROS, which generates telomere dysfunction in the absence of shortening[24]. Dysfunctional telomeres are recognized by γH2AX and 53BP1 localization at telomeres, which are downstream effectors of DDR kinases ATM (ataxia telangiectasia mutated) and ATR (ataxia telangiectasia and Rad3-related)[25,26]. These foci are called TIFs (telomere dysfunction induced foci), TAFs (telomere associated DDR foci), or DDR+ telomeres[27]. Whereas telomere deprotection upon shelterin disruption activates the DDR[26], evidence is lacking that ROS-induced telomere damage is extensive enough to completely displace shelterin. The precise mechanism of ROS-induced DDR activation at telomeres, and whether oxidative modification of telomeric DNA can directly trigger senescence, remains unknown.

Delineating the biological impact of oxidative lesions at telomeres has been challenging because oxidants used to modify DNA have pleiotropic effects on cell signaling, redox status and transcription. To overcome this, we developed and validated a chemoptogenetic tool that produces 8oxoG exclusively at telomeres[28]. This tool uses fluorogen-activating peptides (FAPs) with high affinity for di-iodinated malachite green (MG2I) photosensitizer dye. MG2I generates singlet oxygen ($^1O_2$) upon FAP binding and excitation with far-red light[29]. $^1O_2$ is a main contributor of UVA radiation-induced oxidation reactions, arises from inflammation, lipoxygenases and dioxygenases, and forms primarily 8oxoG when reacting with DNA[30,31]. The physiological importance of 8oxoG is underscored by the evolution of three dedicated enzymes that specifically recognize 8oxoG in various contexts to enable repair and prevent mutations[32,33]. We used a FAP-mCerulean-TRF1 fusion protein to target $^1O_2$ to telomeres[28]. Surprisingly, even repair-deficient cancer cells lacking 8oxoG

[1]Department of Environmental and Occupational Health, University of Pittsburgh School of Public Health, Pittsburgh, PA, USA. [2]UPMC Hillman Cancer Center, Pittsburgh, PA, USA. [3]Department of Pharmacology and Chemical Biology, University of Pittsburgh School of Medicine, Pittsburgh, PA, USA. [4]Departments of Biological Sciences and Chemistry and the Molecular Biosensors and Imaging Center, Carnegie Mellon University, Pittsburgh, PA, USA. [5]Department of Computational and Systems Biology, University of Pittsburgh, Pittsburgh, PA, USA. [6]These authors contributed equally: Mariarosaria de Rosa, Sanjana A. Thosar. ✉e-mail: plo4@pitt.edu

glycosylase (OGG1) are largely unaffected by a single telomeric 8oxoG induction, although repeated inductions over a month causes telomere shortening and instability[28]. However, a role for telomeric 8oxoG in cellular aging could not be delineated in cancer cells.

Here, we demonstrate that in stark contrast to cancer cells, acute production of 8oxoG in telomeres is sufficient to rapidly impair growth of nondiseased human fibroblasts and epithelial cells. Using our chemoptogenetic tool, we show a single 5 min production of telomeric 8oxoG induced numerous hallmarks of cellular senescence within 4 days. Remarkably, even though telomeres are roughly 0.025% of the genome, telomeric 8oxoG rapidly activated ATM and ATR kinases and downstream effectors p53 and p21. Knockout of p53 rescued the growth reduction, indicating that p53 signaling enforces 8oxoG-induced premature senescence. We demonstrate the mechanism is by 8oxoG provoking replication stress-induced DDR activation and telomere fragility, rather than by accelerating telomere losses or shortening. Our data reveal a new mechanism of rapid telomere-driven senescence triggered by a common oxidative stress-induced base lesion that is distinct from 'replicative senescence,' and has important implications for cellular aging linked to oxidative stress.

## Results

**Telomeric 8oxoG initiates rapid senescence in nondiseased cells.**
We showed previously that FAP-TRF1 specifically induces 8oxoG at telomeres when cells are treated with MG2I dye and 660 nm red light together[28]. To understand how nondiseased cells respond to telomeric 8oxoG, we generated clones that homogenously express FAP-mCerulean-TRF1 (termed FAP-TRF1) at telomeres in hTERT, human fibroblast BJ and epithelial RPE1 (retinal pigment epithelial) cell lines (referred to as BJ and RPE FAP-TRF1) (Extended Data Fig. 1a,b). These cells were transduced with telomerase at an early passage, making them amenable for cloning while exhibiting normal karyotypes and DDR pathways.

OGG1 first removes 8oxoG, then APE1 cleaves the backbone and scaffold protein XRCC1 arrives to coordinate repair[32]. To verify 8oxoG formation, we showed increased YFP-XRCC1 colocalization and signal intensity at telomeres after dye and light (DL) that was attenuated in OGG1 knockout cells (ko) or with $^1O_2$ quencher sodium azide (Fig. 1a,b and Extended Data Fig. 1c,e). To compare 8oxoG at telomeres with the bulk genome, we used potassium bromate ($KBrO_3$), which produces primarily genomic 8oxoG, but also damages other cellular components[34]. We showed previously that 40 mM $KBrO_3$ or DL produce similar amounts of telomeric 8oxoGs (around one to five lesions per telomere) in HeLa LT cells[28]. The same 8oxoG detection assay for BJ and RPE FAP-TRF1 cells revealed a dose-dependent increase in 8oxoGs from 5 to 20 min DL, and that 40 mM $KBrO_3$ produced similar amounts of telomere damage (Extended Data Fig. 1f,g). S1 nuclease alone did cleave telomeres, confirming that FAP-TRF1 activation did not immediately induce single-strand breaks.

We investigated how telomeric 8oxoG impacts cell growth. Treating BJ and RPE FAP-TRF1 cells for 5 min with DL, but not dye or light alone, significantly reduced cell growth just 4 days after treatment (Fig. 1c,d and Extended Data Fig. 2a). The extent of growth reduction depended on the light duration, showing that the cellular response was proportional to the amount of telomeric damage (Extended Data Fig. 2b). Parental hTERT BJ and RPE cells lacking FAP-TRF1, and FAP-TRF1-expressing HeLa and U2OS cancer cells, showed no growth changes after DL treatment (Extended Data Fig. 2c–e). These data confirm that growth reduction in nondiseased cells requires FAP-TRF1, and that cancer cells are insensitive. DL exposure of nonclonal primary BJ cells expressing variable FAP-TRF1 levels also reduced growth (Fig. 1e and Extended Data Fig. 2f), indicating that reduction occurs regardless of telomerase status. Interestingly, 2.5 mM $KBrO_3$ for 1 h reduced cell growth to levels comparable with 5 mins DL (compare Fig. 1c,d with Extended Data Fig. 2g,h). These data demonstrate that nondiseased cells are highly sensitive to elevated 8oxoG at the telomeres, although telomeres are a tiny fraction of the genome.

Next, we asked whether the growth reduction was due to senescence, characterized by persistent growth arrest and various other phenotypes depending on the cell type and mechanism of senescence induction[5]. Consistent with impaired growth as early as 24 h after DL treatment, we observed a reduction in EdU-positive S-phase cells (Fig. 1f and Extended Data Fig. 2i,j). These changes were comparable with those seen after 2.5 mM $KBrO_3$ treatment, while 10 mM $KBrO_3$ and 20 J m$^{-2}$ UVC dramatically reduced S-phase cells. DL reduced RPE FAP-TRF1 cell colony formation, and increased senescence-associated β-galactosidase (SA-β-gal)-positive BJ FAP-TRF1 cells 4 days after exposure, whereas dye or light alone did not (Fig. 1g–h,j and Extended Data Fig. 2k). DL also increased nuclear area—a morphological change associated with senescence (Extended Data Fig. 2l). 2.5 mM $KBrO_3$ induced an increase in SA-β-gal staining identical to that induced by 5 min DL, consistent with the similar growth reductions (Extended Data Fig. 2g,h). DL for 20 min dramatically increased SA-β-gal staining and reduced colony formation, similar to the genotoxic control etoposide (ETP), and consistent with greater growth inhibition (Fig. 1g,i and Extended Data Fig. 2g).

Senescent cells remain metabolically active despite their nonproliferative state[35]. Mitochondria oxygen consumption rate (OCR) measured after DL revealed slight increases in the basal OCR (Fig. 1k). Treatment with the mitochondrial uncoupler FCCP (Trifluoromethoxy carbonylcyanide phenylhydrazone) dramatically increased the maximal respiration of DL-treated cells until the mitochondria were inhibited with rotenone. Our results are consistent with previous reports of elevated OCR in senescent cells[36,37]. In summary, our data show that human fibroblasts and epithelial cells undergo rapid, premature senescence following telomeric 8oxoG formation.

**Fig. 1 | Acute telomeric 8oxoG initiates rapid premature senescence. a**, YFP-XRCC1 localization to telomeres indicated by FAP-mCer-TRF1 after 10 min dye + light (DL) treatment of RPE FAP-TRF1 cells. **b**, Percent YFP-XRCC1 positive telomeres per nucleus after no treatment (UT) or 10 min DL in wild-type or OGG1ko RPE FAP-TRF1 cells. Error bars represent the mean ± s.d. from the indicated number n of nuclei analyzed from a representative experiment. Statistical analysis by one-way ANOVA (***$P < 0.001$). Immunoblot for FAP-TRF1 and OGG1 in extracts from RPE FAP-TRF1 wild-type and OGG1ko cells. Arrow indicates nonspecific band stained by anti-OGG1. **c–e**, Cell counts of BJ (**c**), RPE (**d**) or primary BJ (**e**) FAP-TRF1 cells obtained 4 days after recovery from 5 or 20 min dye (D) and light (L) alone, or in combination (DL) as indicated, relative to untreated cells. **f**, RPE FAP-TRF1 cell cycle analysis 24 h after no treatment or 5 min D, L, DL, 20 J m$^{-2}$ UVC, or 1 h with 2.5 or 10 mM $KBrO_3$ determined by flow cytometry. **g**, RPE FAP-TRF1 colony formation efficiency 7–10 days after indicted treatment. **h,i**, Percent β-galactosidase-positive BJ FAP-TRF1 cells obtained 4 days after the indicated treatments; 2.5 mM $KBrO_3$ and 50 μM ETP treatments were for 1 h. In **c–i**, error bars represent the mean ± s.d. from the number of independent experiments indicated by the black circles. Statistical significance was determined by one-way ANOVA (ns, not significant; *$P < 0.05$; **$P < 0.01$; ***$P < 0.001$; ****$P < 0.0001$). **j**, Representative image of 5 min DL-treated BJ FAP-TRF1 β-galactosidase-positive cells. Arrows mark positive cells (turquoise). **k**, Mitochondrial respiration was examined 24, 48 and 96 h after 5 min D, L or DL. Data are means and error bars are ±95% CI from two independent experiments with seven to eight technical replicates each for BJ and RPE FAP-TRF1 cells.

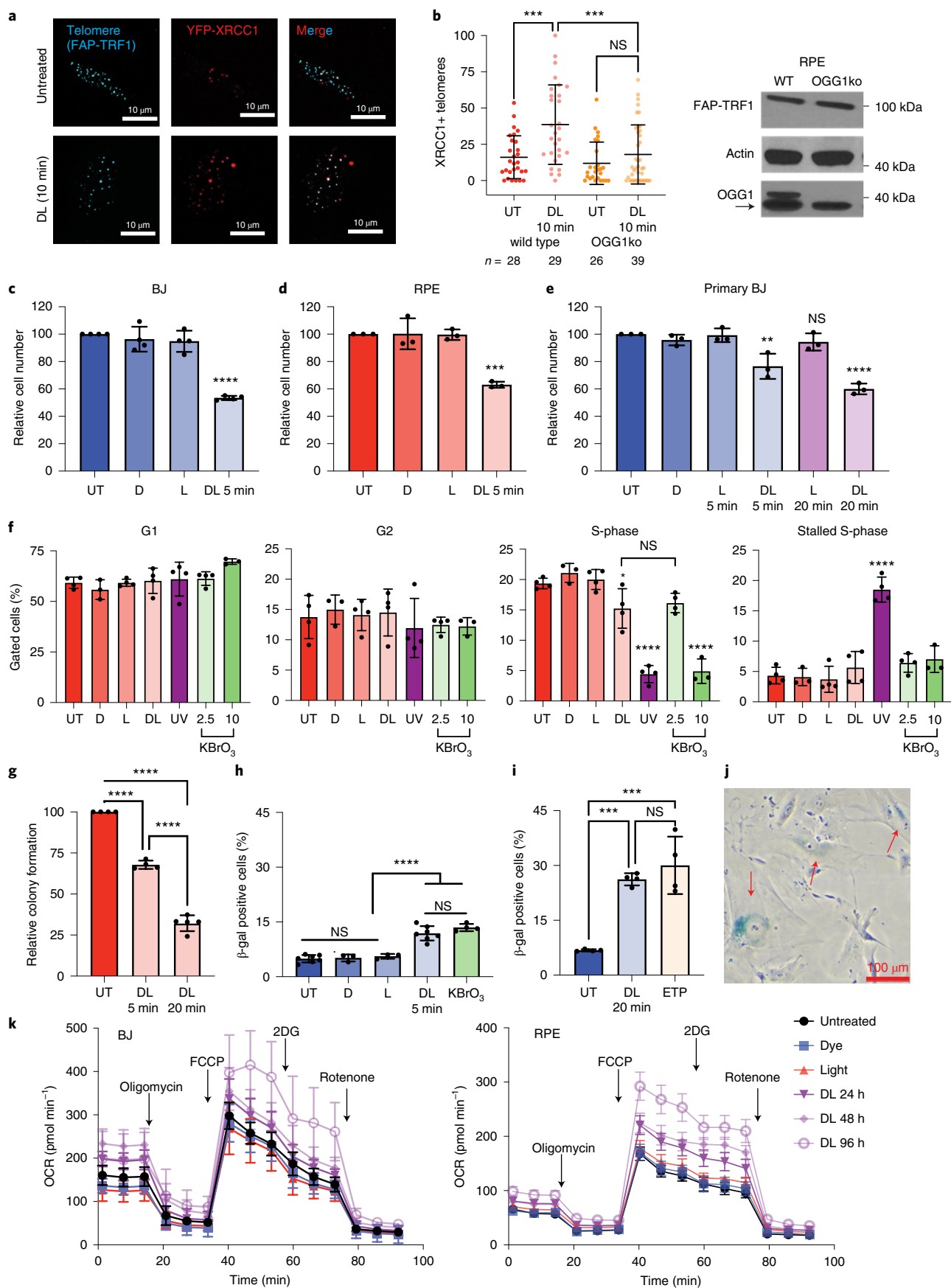

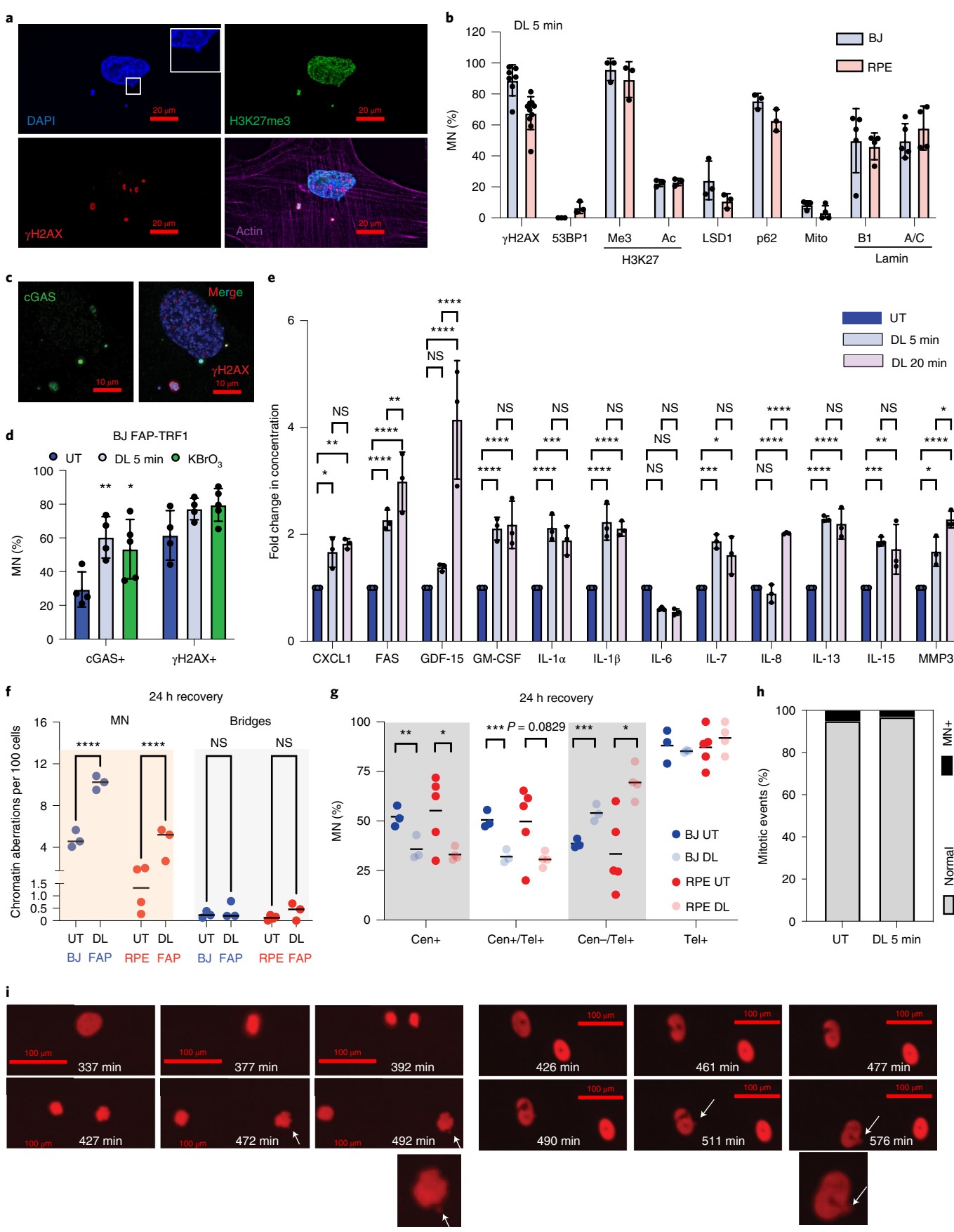

**Fig. 2 | Telomeric 8oxoG production increases cytoplasmic DNA. a**, Image of γH2AX, H3K27me3 and actin in BJ FAP-TRF1 cells 4 days after 5 min DL. Inset, enlargement of blebbing MN. **b**, Percentage of MN positive for the indicated markers from BJ and RPE FAP-TRF1 cells 4 days after 5 min DL. Error bars represent the mean ± s.d. from the number of independent experiments indicated by black circles. **c**, BJ FAP-TRF1 cells stained for cGAS and γH2AX 4 days after 5 min DL treatment. **d**, Percentage of MN that are cGAS or γH2AX positive 4 days after 5 min DL or 1 h 2.5 mM KBrO$_3$ treatment as in panel **c**. **e**, SASP analysis of BJ FAP-TRF1 cells 7 days post-treatment with 5 or 20 min DL. Concentration normalized to the final cell number in each sample. Data are presented as fold changes. Actual concentrations are in Supplementary Table 1. **f**, Quantification of MN and chromatin bridges 24 h after DL as visualized by DAPI. At least 500 cells were counted per experiment. **g**, Quantification of MN from panel **f** showing the percentage of MN positive for centromeric (Cen+) or telomeric (Tel+) DNA in total, and the percentage positive for both (Cen+/Tel+) or only telomeric DNA (Cen–/Tel+). At least 30 MN were analyzed for each experiment. For panels **d**–**g**, error bars represent the mean ± s.d. from the number of independent experiments indicated by the black (**d**,**e**) or red and blue (**f**,**g**) circles. Statistical significance for panels **d**–**f** determined by two-way ANOVA, and for panel **g** by multiple *t*-tests (ns, not significant; *$P < 0.05$; **$P < 0.01$; ***$P < 0.001$; ****$P < 0.0001$). **h**, Percentage of mitoses resulting in a MN from live imaging of BJ FAP-TRF1 cells 24 h following 5 min DL. For UT $n = 60$ and DL 5 min $n = 64$ mitoses observed from two independent experiments. **i**, Stills from live-cell imaging. Left panel, a mitosis that produced a MN (arrow); right panel, an interphase cell with nuclear blebbing (arrow).

**Telomeric 8oxoG increases cytoplasmic DNA.** A shared hallmark of senescence and cancer is increased micronuclei (MN), also termed cytoplasmic chromatin fragments (CCF) in senescent cells, which can arise by different mechanisms[5,38,39]. DL increased MN in BJ and RPE FAP-TRF1 cells 4 days after exposure (Extended Data Fig. 3a). Consistent with CCFs, the MN from treated cells localized within the cytoplasm, stained positive for γH2AX, heterochromatin marker H3K27Me3 and autophagy marker p62 and negative for 53BP1 and euchromatin markers LSD1 (lysine-specific histone demethylase 1 A) and H3K27Ac and for mitochondria (Fig. 2a,b and Extended Data Fig. 3b)[39]. Lamin B1 encapsulates CCFs[40], and nearly 50% of the MN were positive for Lamin B1 and Lamin A/C, although telomere damage decreased overall Lamin B1 expression (Fig. 2b and Extended Data Fig. 3b–d), which is a senescence hallmark[5].

MN are sensed by the cytoplasmic DNA sensor cGAS, which promotes the senescence-associated secretory phenotype (SASP)[41]. DL or KBrO$_3$ increased the percentage of MN positive for cGAS (Fig. 2c,d). Cells displaying cGAS+ MN also showed increased nuclear γH2AX, indicating that they were responding to DNA damage (Extended Data Fig. 3e). DL increased common SASP factors, including GDF-15, FAS and IL-1β, compared with untreated cells, 7 days after recovery (Fig. 2e). Positive controls of 10 mM KBrO$_3$ and ETP also produced a robust SASP, which was greater than just damaging telomeres (Supplementary Table 1).

To test whether the MN arise from chromosomal breakage-fusion-bridge (BFB) cycles and lagging chromosomes[10], we quantified chromatin bridges 24 h after telomeric 8oxoG induction. While MN increased significantly, chromatin bridges did not (Fig. 2f). Moreover, the percentage of MN positive for centromere DNA decreased, while MN negative for centromere DNA, but positive for telomere DNA, increased (Fig. 2g and Extended Data Fig. 3f). Thus, telomeric 8oxoG does not increase lagging chromosomes (Cen+/Tel+ MN), but rather increases acentric fragments (Cen–/Tel+ MN), which can arise outside of mitosis[42,43]. Senescent cells can

produce MN by chromatin blebbing in interphase instead of by mitotis[39,40]. Live-cell imaging of BJ FAP-TRF1 cells expressing H2B-RFP showed no change in the percentage of mitoses giving rise to MN 24 h after DL (Fig. 2h and Supplementary Videos 1 and 2). This confirms that, unlike shelterin disruption, acute telomere 8oxoG damage does not induce BFB[44]. However, while difficult to quantify, we observed nuclear DNA blebbing from the primary nucleus forming MN after telomere damage, consistent with the mechanism of CCF formation associated with senescence (Fig. 2i and Supplementary Videos 3 and 4).

Since apoptotic cells induce DNA breaks, which can form MN, we tested for apoptosis by Annexin V (AV) and propidium iodine (PI) staining. While the positive controls of 20 J m$^{-2}$ UV or 10 mM KBrO$_3$ induced late apoptotic (AV+/PI+) and dead (AV–/PI+) cells, DL did not increase cell death or apoptosis (Extended Data Fig. 4a,b). Furthermore, DNA breaks were not induced immediately or 24 h after DL, in contrast to the H$_2$O$_2$ positive control (Extended Data Fig. 4c). In summary, $^1$O$_2$ induction at telomeres does not induce DNA breaks or apoptosis directly, but instead increases cytoplasmic DNA in a manner consistent with senescence.

**p53 DNA damage signaling triggers 8oxoG induced senescence.** DDR signaling drives cell cycle arrest and growth inhibition leading to senescence if the damage is extensive or unresolved[5]. DL activated the ATM/Chk2 pathway within minutes (Fig. 3a), which is striking because small base modifications are not canonically associated with ATM activation[45,46]. Treating cells with ATM inhibitor (ATMi) after DL partially rescued the damage-induced colony formation and β-gal phenotypes (Fig. 3b,c), confirming the role of ATM role in telomeric 8oxoG-induced senescence. Tumor suppressor p53 is downstream of ATM/Chk2 and drives the transcription of numerous DNA repair factors, the cell cycle checkpoint and senescence enforcement[47]. Shortly after DL, the p53 antagonist MDM2 was degraded, causing p53 protein stabilization and induction of p21—a p53 target protein (Fig. 3d,e). Activation of p53 and p21 prevents

**Fig. 3 | p53 DNA damage signaling is required for 8oxoG-induced senescence. a**, Immunoblots of phosphorylated ATM (pATM) and phosphorylated Chk2 (pChk2) at the indicated recovery time following 5 min DL. **b**, RPE FAP-TRF1 colony formation after 7 days recovery from DL. Cells were cultured with ATMi KU60019 or DMSO only during recovery. Numbers are relative to untreated DMSO control. **c**, Percentage of β-gal positive BJ FAP-TRF1 cells obtained 4 days after DL. Cells were cultured with ATMi KU60019 or DMSO only during recovery. **d**, Schematic of canonical DNA damage-induced p53 activation by ATM and ATR kinases. Created with Biorender.com. **e**, Immunoblot of untreated BJ FAP-TRF1 cells, or treated with DL and recovered for the indicated times. UV = 20 J m$^{-2}$ UVC. ETP = 1 h 50 μM ETP. **f**, Heat map of mRNASeq results from FAP-TRF1-expressing RPE, BJ and HeLa cells 24 h after no treatment (NT) or 5 min DL. Shown are the top altered genes and p53 target genes are in red. Each column is an independent replicate. **g**, Cell counts of wild-type, p16ko, p53ko or p16+p53 double ko of BJ (blue) or RPE (red) FAP-TRF1 cells 4 days after recovery from 5 min DL relative to respective untreated cells. Immunoblot below shows FAP-TRF1, p53, p16 expression. **h**, RPE FAP-TRF1 colony formation after 7 days recovery from DL. **i**, Percentage of β-gal positive BJ FAP-TRF1 cells obtained 4 days after treatment. **j**, Percent EdU-positive cells observed 24 h after 5 min DL (light red or light blue). Over 200 cells were scored per condition in each experiment. For **b**,**c**, and **g**–**j**, error bars represent the mean ± s.d. from the number of independent experiments indicated by the black circles. Statistical significance in **b**,**c** and **h**–**j** determined by two-way ANOVA, and for **g** by one-way ANOVA (ns, not significant; *$P < 0.05$; **$P < 0.01$; ***$P < 0.001$; ****$P < 0.0001$).

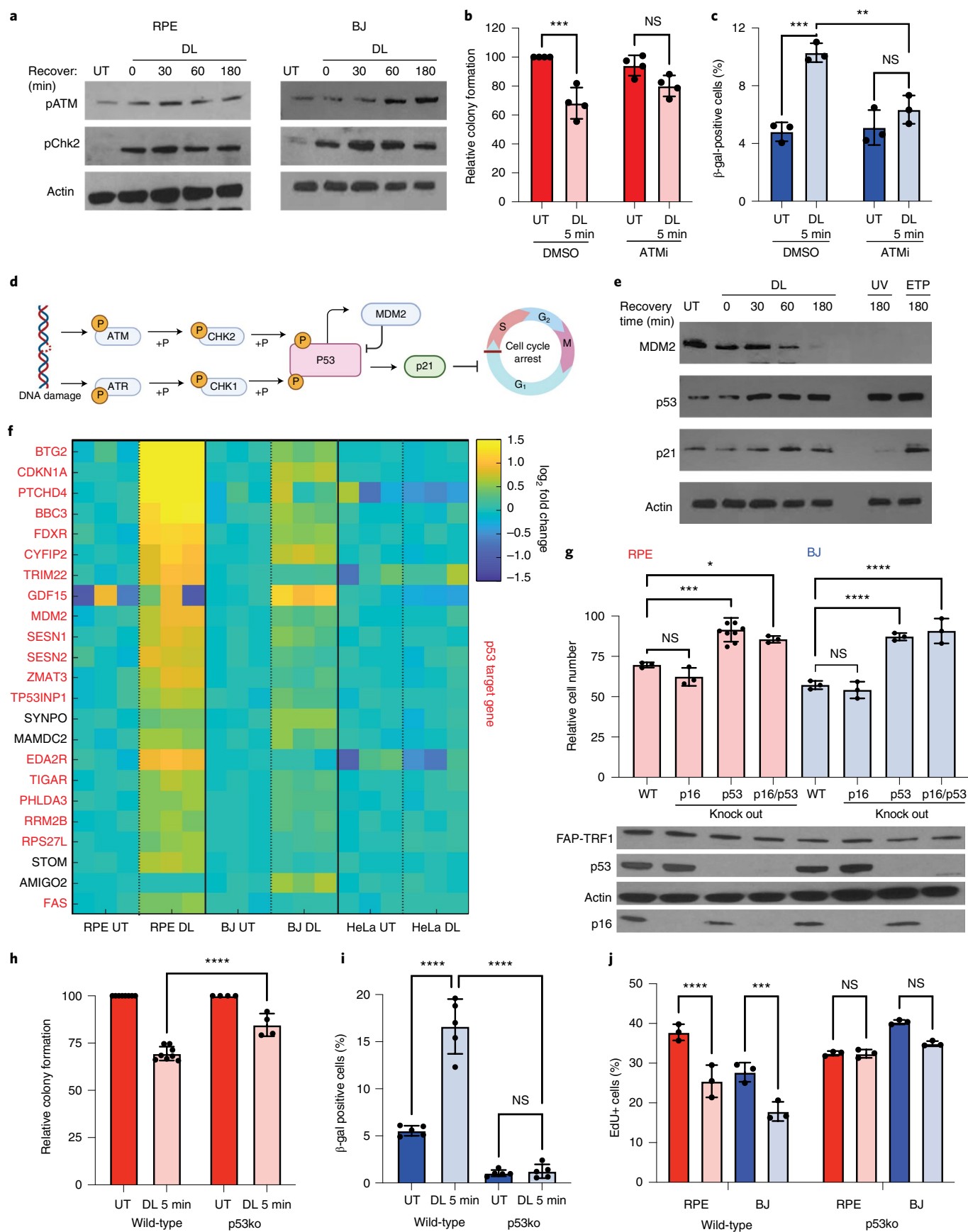

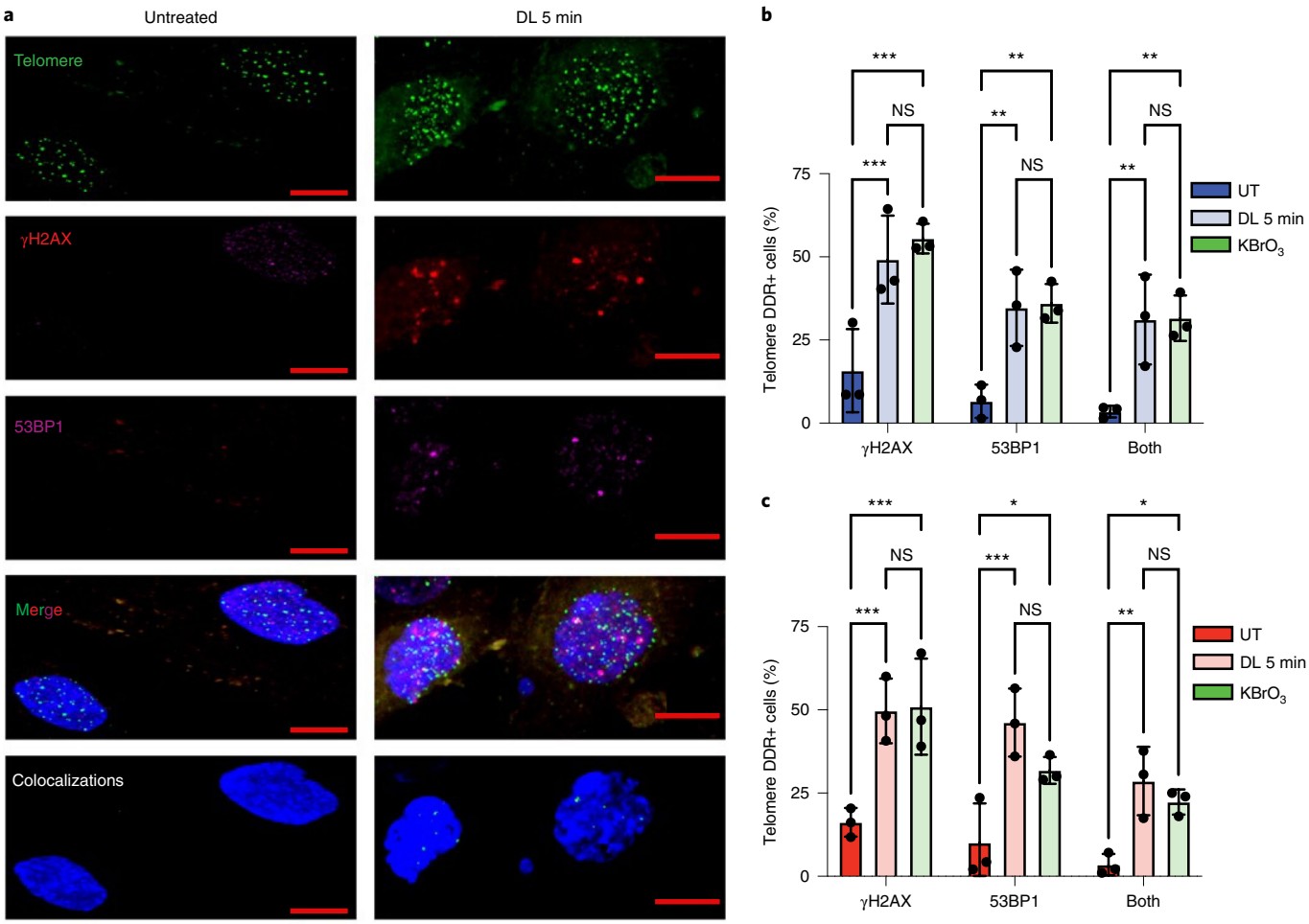

**Fig. 4 | Telomeric 8oxoG promotes a localized DDR. a**, Representative IF images showing γH2AX (red) and 53BP1 (purple) staining with telomeres (green) by telo-FISH for BJ FAP-TRF1 cells 24 h after no treatment or 5 min DL. Colocalizations panel shows NIS-Elements-defined intersections between 53BP1 and/or γH2AX with telomeres. Scale bars, 10 μm. **b,c**, Quantification of percentage of cells exhibiting telomere foci colocalized with γH2AX, 53BP1 or both for BJ (**b**) and for RPE (**c**) FAP-TRF1 cells 24 h after 5 min DL or 2.5 mM KBrO₃ treatment. Error bars represent the mean ± s.d. from three independent experiments in which more than 50 nuclei were analyzed per condition for each experiment. Statistical significance determined by two-way ANOVA (ns, not significant; *$P < 0.05$; **$P < 0.01$; ***$P < 0.001$).

transcription of S-phase factors by reducing RB phosphorylation and inhibiting E2F transcription factors, which occurred following telomeric 8oxoG induction (Extended Data Fig. 5a). Consistent with ATM activating p53 in response to telomeric 8oxoG, cells treated with ATMi after DL showed attenuated p53 induction (Extended Data Fig. 5b,c).

Next, we examined the transcriptional response to telomeric 8oxoG 24 h after DL. HeLa cells showed no significant changes after acute telomeric 8oxoG induction (Fig. 3f), consistent with the lack of growth changes[28]. In contrast, RPE and BJ FAP-TRF1 cells showed significant gene expression changes after telomeric 8oxoG, which were not proximal to the telomeres, demonstrating that these changes were not an artifact of inducing damage at the telomeres (Fig. 3f and Extended Data Fig. 5d–g). The Hallmark gene set enrichment analysis revealed downregulation of replication and cell cycle pathways consistent with senescence (Supplementary Table 2), and p53 pathway upregulation, consistent with upregulation of p53 target genes after treatment (Fig. 3f)[48].

Both p53 and p16 drive senescence and reduce RB phosphorylation[49]. However, p16ko did not rescue the DL-induced growth reduction, while p53ko alone or in combination with p16 did (Fig. 3g). Compared with wild-type cells, p53ko cells displayed an attenuated

reduction in growth as a function of light duration (Extended Data Fig. 5h). Furthermore, p53 loss suppressed the reduction in colony formation, increase in SA-β-gal and reduction in EdU incorporation in treated cells (Fig. 3h–j), and rescued KBrO₃ induced growth reduction (Extended Data Fig. 5i). Consistent with p16ko failing to rescue senescence, DL did not increase p16 mRNA, while 10 mM KBrO₃ did (Extended Data Fig. 5j). We also observed upregulation of p21 mRNA, which was sustained up to 4 days post-treatment (Fig. 3e,f and Extended Data Fig. 5k). At 24 h after treatment, p21 was induced only in EdU negative, nonreplicating and/or senescent wild-type cells, but not in p53ko cells (Extended Data Fig. 6).

Given the p53 requirement, we reasoned that cells in which telomeric 8oxoG triggered a DDR were more likely to activate p53 and, thus, senesce. Consistent with this, DL dramatically increased DDR factor 53BP1 in p53 positive cells, compared with p53 negative cells (Extended Data Fig. 5l,m). As a control, MDM2 antagonist Nutlin induced a greater fraction of p53 positive cells but did not induce 53BP1. These data indicate that cells that experienced a greater telomeric 8oxoG-induced DDR also showed p53 activation. In summary, these results demonstrate that telomeric 8oxoG is sufficient to trigger a DDR and activate p53 and p21, which drives premature senescence.

**8oxoG promotes a localized telomeric DDR.** The striking DDR and p53 activation observed following targeted 8oxoG formation probably emanated from localized DDR activation at telomeres. We tested for γH2AX and 53BP1 recruitment to telomeres in interphase cells 24 h after DL. Treatment increased the percentage of cells with one or more DDR-positive telomeres, and dramatically increased (around tenfold) cells showing telomeres colocalized with both DDR markers (Fig. 4a–c). Binning this data revealed significant increases in cells displaying one to three or four or more DDR+ telomeres (Extended Data Fig. 7a,b). Previous studies showed four to five γH2AX+ telomeres predicts replicative senescence in human fibroblasts[50]. Summing the percentage of cells with four or more telomeres positive for γH2AX or 53BP1 yielded 20–30% for DL-treated cells and only 2% for untreated cells (Extended Data Fig. 7c). The telomere DDR after DL was comparable with DDR after 2.5 mM KBrO₃, even though this oxidant damages the entire cell (Fig. 4b,c). Moreover, this dose of KBrO₃ produces a similar increase in senescent cells as 5 min DL (Fig. 1g,h and Extended Data Fig. 2g,h). While the percentage of cells with DDR+ telomeres decreased after 4 days, it remained higher than background for both treatments, indicating persistent telomeric DDR (Extended Data Fig. 7d,e). These observations confirm that telomeres are hotspots for oxidative damage, and suggest that telomeres are prone to acute and persistent DDR activation upon 8oxoG processing.

Next, we confirmed telomeric DDR by γH2AX staining of metaphase chromosomes (meta-TIF). The average number of chromatids staining positive for both γH2AX and telomere PNA (peptide nucleic acid) was 4.4 per metaphase, and positive for γH2AX but negative for telomere PNA was 0.9 per metaphase after treatment (Extended Data Fig. 7f,g). This suggests that most 8oxoG-induced DDR was not due to telomere loss. We also analyzed the distribution of γH2AX foci at chromatid ends versus internal sites, since chromosome ends missing a telomere are undetectable in interphase cells[51]. While 60% of γH2AX foci localized to chromatid ends in untreated cells, consistent with telomeres as damage hotspots, this increased to 83% after DL (Extended Data Fig. 7h).

**8oxoG disrupts telomere replication without causing shortening.** Next, we investigated the mechanism for 8oxoG-induced DDR activation at telomeres. Telomeric 8oxoG triggered senescence by 4 days, a timeframe typically insufficient to observe notable telomere shortening, particularly in telomerase-proficient cells (normally requires weeks). Analysis of telomere restriction fragments revealed no change in the bulk telomere lengths 4 days after DL (Extended Data Fig. 8a). Since a few critically short telomeres can promote senescence[52], we used the telomere shortest length assay (TeSLA) to visualize the shortest individual telomeres. Although we detected individual telomeres much shorter than in the bulk population, DL did not increase the percentage of short or truncated telomeres (Extended Data Fig. 8b). Thus, telomere shortening does not need to precede oxidative stress-induced senescence.

We next examined telomere integrity by telo-FISH on metaphase chromosomes in p53ko cells to ensure damaged cells could progress to mitosis. Chromatid ends were scored as showing one telomeric foci (normal), multiple foci (fragile) or no staining (signal-free end) (Fig. 5a,b). DL induced little-to-no change in signal-free ends representing lost or undetectable telomeres, or in dicentric chromosomes representing chromosome fusions (Fig. 5c,d and Extended Data Fig. 8c). Consistent with a lack of telomere losses, we also observed no reduction in telomere foci 4 days after DL in wild-type interphase cells (Extended Data Fig. 8d). However, DL significantly increased fragile telomeres 24 h after treatment (Fig. 5e,f).

Since shelterin disruption can activate ATM and induce senescence[53,54] and 8oxoG can disrupt TRF1 and TRF2 binding in vitro[55], we examined whether DL reduced shelterin at telomeres. TRF2 or TRF1 deletion generates DDR+ telomeres by causing deprotection and fusions or telomere fragility, respectively[54]. TRF1 deletion also induces growth arrest and SA-β-gal, which is rescued by p53 inhibition[54,56]. Although TRF1 deletion is not physiologic, these phenotypes are strikingly similar to those observed with telomeric 8oxoG formation. However, DL failed to induce loss of FAP-mCER-TRF1 at telomeres, as evidenced by no change in mCerulean foci number and signal intensity (Extended Data Figs. 1d and 8e). TRF2 staining revealed no loss of TRF2 in general, or at γH2AX positive telomeres, consistent with no fusions (Extended Data Fig. 8c,f,g). Thus, telomeric 8oxoG induces premature senescence without telomere shortening and losses or deprotection via shelterin disruption, but rather induces telomere fragility.

Since fragile telomeres are associated with replication stress[54,56,57], we also tested for mitotic DNA synthesis (MiDAS), which can occur at difficult-to-replicate regions to enable completion of DNA synthesis, and is detected by EdU incorporation during mitosis[58]. RPE FAP-TRF1 cells were treated with DL, recovered, treated with CDK1inhibitor R0-3306 to synchronize in G2, then released into medium containing EdU and colcemid to visualize DNA synthesis in metaphase. DL induced at least one telomere MiDAS event in 79% of treated cells, compared with only 39% in untreated cells (Fig. 5g). Telomere MiDAS occurs primarily by conservative DNA synthesis on a single chromatid, consistent with break-induced-replication (BIR), in contrast to homologous recombination (HR), which requires semiconservative synthesis on both chromatids (Fig. 5h)[59,60]. Whereas untreated cells displayed an average of 0.2–0.3 telomere MiDAS events per metaphase (single or both chromatid; median 0), DL-treated cells showed a significant increase in single chromatid telomere MiDAS (average and median of one per metaphase) (Fig. 5i). These single chromatid events almost exclusively stained positive for telomere PNA, consistent with BIR. These data indicate that acute telomeric 8oxoG formation triggers mitotic DNA synthesis, suggesting that the lesions prevented the completion of telomere replication in S-phase.

**Replication promotes telomeric 8oxoG-induced DDR.** To test whether S-phase cells are more sensitive to telomeric 8oxoG, we

**Fig. 5 | 8oxoG directly disrupts telomere replication. a,b**, Representative images of telo-FISH staining of metaphase chromosomes from BJ FAP-TRF1 p53ko cells 24 h after no treatment (**a**) or 5 min DL **b**, Images were scored for telomeric signal-free ends (yellow arrowheads) and fragile telomeres (green arrowheads). Green foci are telomeres and pink foci are CENPB centromeres. Scale bars, 10 μm. **c,d**, The number of telomeric signal-free chromatid ends per metaphase in BJ (**c**) and RPE (**d**) FAP-TRF1 cells. **e,f**, The number fragile telomeres per metaphase in BJ (**e**) and RPE (**f**) FAP-TRF1 cells. For **c–f**, error bars represent mean ± s.d. from n = 71 (UT) and 72 (DL 5 min) for BJ and n = 61 (UT) and 63 (DL 5 min) for RPE, metaphases analyzed from three independent experiments, normalized to the chromosome number. Statistical analysis by two-tailed Mann–Whitney (ns, not significant; **P < 0.01; ****P < 0.0001). **g**, Schematic of MiDAS experiment in p53ko RPE FAP-TRF1 cells (Top) and representative metaphase spread with Telo PNA (green) and EdU staining (red). Arrows point to telomeric MiDAS. Scale bars, 10 μm. **h**, Schematics for EdU events at a single chromatid (BIR) and both chromatids (HR). Representative images from DL-treated RPE FAP-TRF1 cells are shown below. **i**, Telomere MiDAS events at a single chromatid (left) or both chromatids (right). Events are scored for chromatid ends staining positive (Telo+) or negative (Telo−) for telomeric PNA. Error bars represent the mean ± s.d. from 51 (UT) and 61 (DL 10 min) metaphases from two independent experiments. Statistical analysis by one-way ANOVA (ns, not significant; ****P < 0.0001).

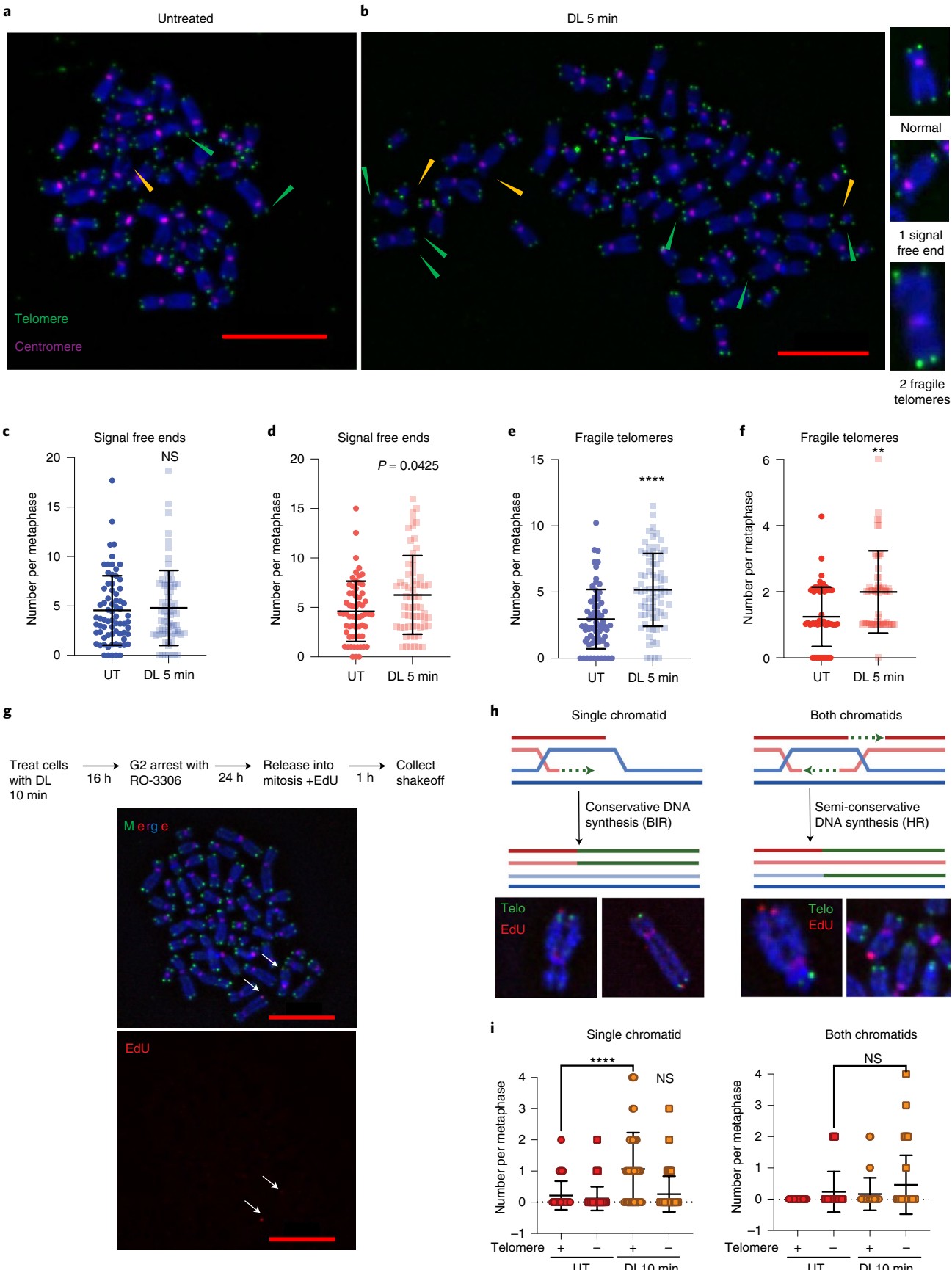

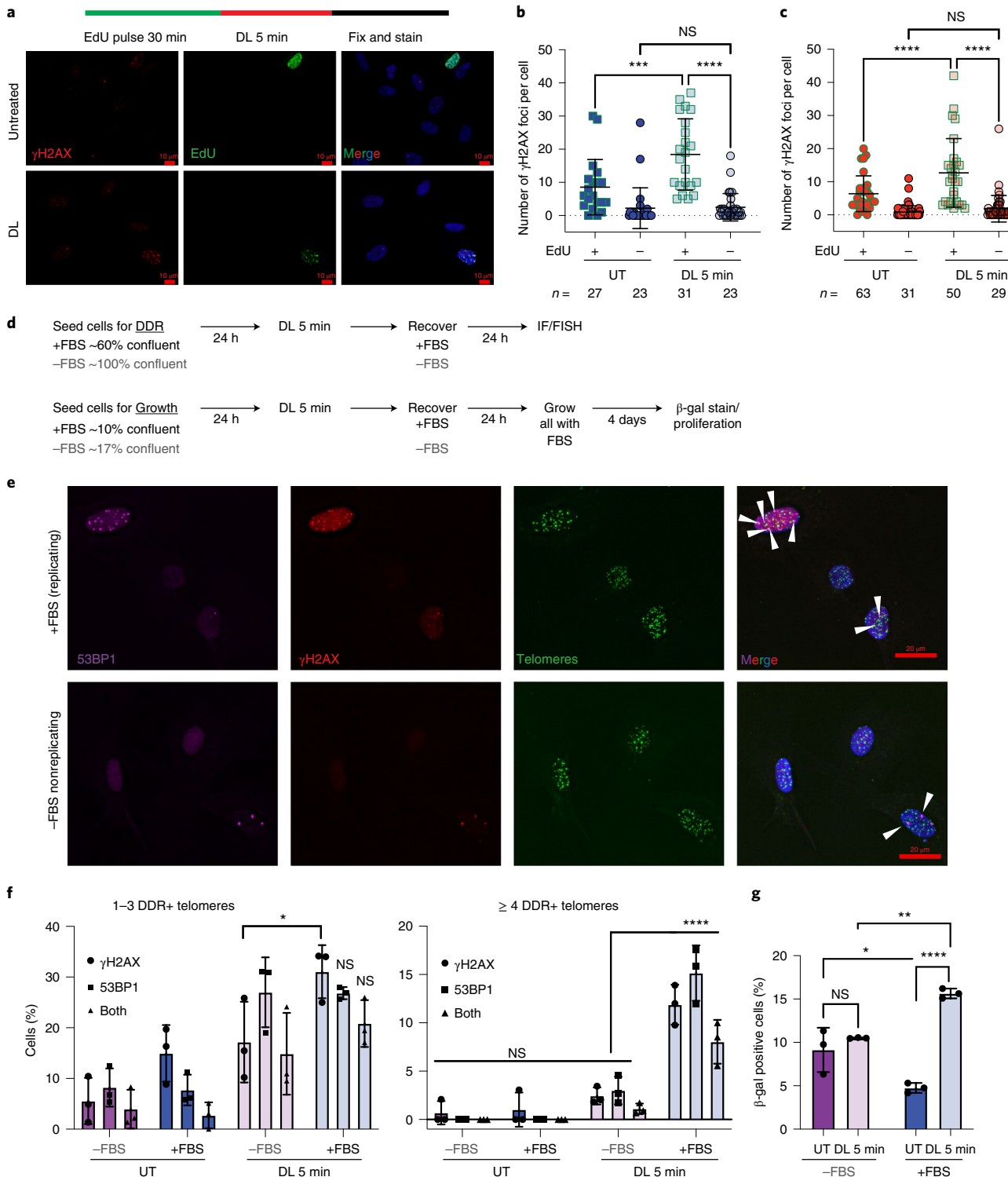

**Fig. 6 | Replicating cells are more sensitive to telomeric 8oxoG. a**, Schematic shows experiment for EdU labeling of S-phase cells. Representative image of γH2AX (red) and EdU (green) staining of BJ FAP-TRF1 cells after 0 h recovery from 5 min DL. Scale bar, 10 μm. **b,c**, Number of γH2AX foci per EdU– and EdU+ cells for BJ (**b**) and RPE (**c**) FAP-TRF1 cells. Error bars represent the mean ± s.d. from the indicated *n* number of nuclei analyzed from two independent experiments. Statistical analysis by one-way ANOVA (ns, not significant; ***$P < 0.001$; ****$P < 0.0001$). **d**, Schematic of experiments for telomere DDR detection (top, **e,f**) and senescence assays (β-gal and proliferation) (bottom, **g** and Extended Data Fig. 10d) in replicating (cells grown with 10% FBS (+FBS)) and nonreplicating (cells grown with 0.1% FBS (–FBS)) BJ FAP-TRF1 cells. **e**, Representative IF/FISH images for the telomere DDR experiment. Scale bar, 20 μm. **f**, Percentage of cells with one to three or four or more DDR+ (γH2AX, 53BP1, or both) telomeres from **e**. Over 70 nuclei were analyzed per condition per experiment. **g**, Cells were seeded in medium with 0.1% (−) or 10% (+) FBS, treated the next day with 5 min DL and then recovered 24 h with 0.1 (−) or 10% (+) FBS. All cells were cultured in 10% FBS medium another 4 days before staining for β-gal activity. At least 300 cells were analyzed per condition per experiment. For **f,g**, error bars represent the mean ± s.d. from the number of independent experiments indicated by the black circles. Statistical significance determined by two-way ANOVA (ns, not significant; *$P < 0.05$; **$P < 0.01$; ****$P < 0.0001$).

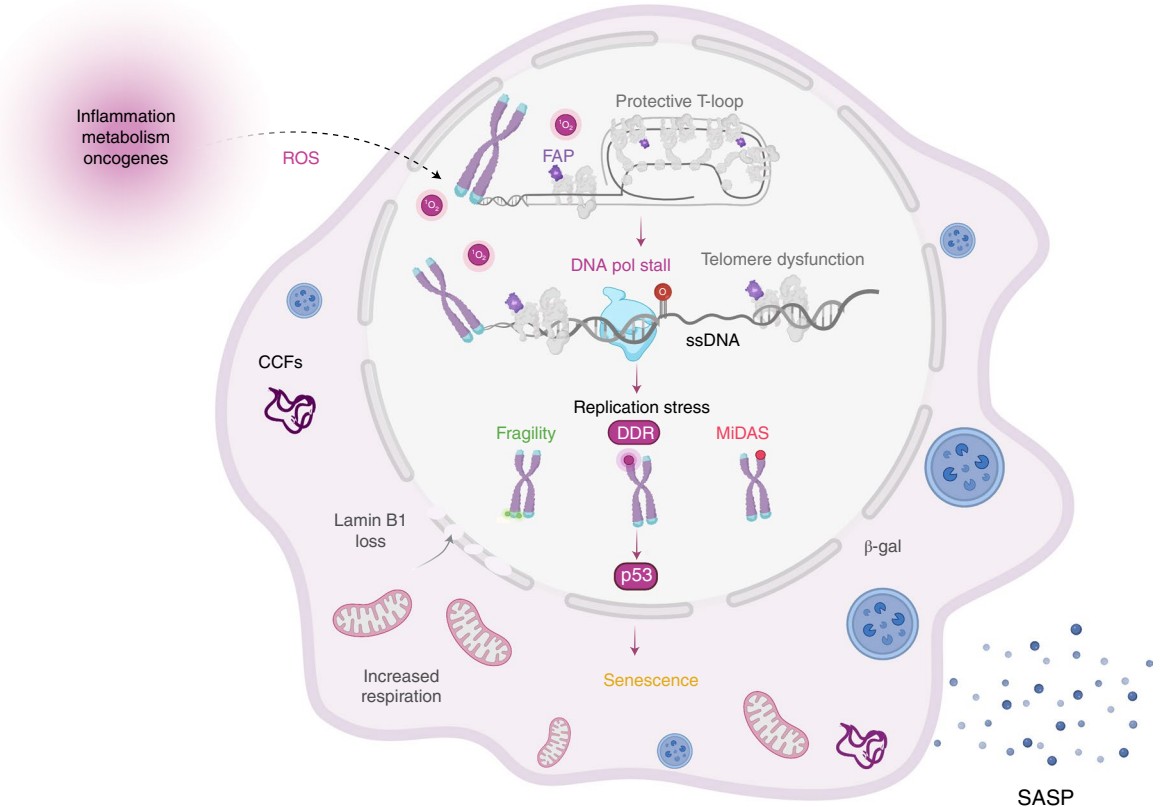

**Fig. 7 | Model for telomere 8oxoG induced senescence.** Following induction of ROS by endogenous or exogenous stressors, telomeres are susceptible to oxidative DNA damage, including formation of the common lesion 8oxoG. When a replication fork encounters 8oxoG in the telomere, it may stall, resulting in excess ssDNA, leading to replication stress. Replication stress can lead to telomere fragility, localized DDR signaling and MiDAS repair at the telomere. Telomere DDR results in p53 activation, which promotes cellular senescence including multiple characteristic hallmarks. Imagine created with BioRender.com.

prelabeled replicating cells with EdU before DL, then immediately stained for DDR (Fig. 6a). Telomeric 8oxoG significantly increased γH2AX foci only in cells that were replicating during the treatment (Fig. 6b,c). Consistent with this, the ATR/Chk1 replication stress response was activated immediately after telomeric 8oxoG induction, and we observed a significant increase in nuclear γH2AX signal intensity in EdU+, but not EdU−, cells 1 h after treatment (Extended Data Figs. 9a–c). The signal decreased 3–12 h after treatment, but increased again in EdU+ cells at 24 h, compared with EdU− cells. A similar second wave of DDR was reported following H2O2 treatment, and was proposed to result from increased replication fork encounters with DNA lesions or repair intermediates[61]. We also observed this trend of immediate DDR activation, reduction and rebound at telomeres (Extended Data Figs. 9d–f).

Next, we investigated the role of DNA replication in telomeric 8oxoG-induced DDR and premature senescence. Fibroblasts synchronize to G0/G1 when serum starved and confluent. We seeded near confluent or subconfluent BJ FAP-TRF1 cells with 0.1% FBS (−FBS) or 10% FBS (+FBS), respectively, treated with DL and recovered in −FBS or +FBS medium (Fig. 6d upper panel). Quiescence was confirmed by a reduction in EdU incorporation and cyclin A expression (Extended Data Figs. 10a–c). While telomeric 8oxoG increased both cells with one to three and four or more DDR+ telomeres in replicating (+FBS) cultures, as expected, the treatment only increased cells with one to three, but not four or more, DDR+ telomeres in quiescent cultures (−FBS) (Fig. 6e,f). Because four or more DDR+ telomeres predicts senescence[50], we tested whether preventing DNA replication for 24 h after treatment would rescue senescence. Quiescent cells treated and recovered in 0.1%

FBS, before culturing in 10% FBS, displayed no increase in β-gal positive cells and showed attenuated growth reduction compared with treated replicating cells (Fig. 6d lower panel, Fig. 6g and Extended Data Fig. 10d). In contrast to proliferating cells, quiescent cells showed attenuated DDR and p53 signaling (Extended Data Fig. 10e). Collectively, our data show that telomeric 8oxoG promotes both replication and p53-dependent senescence in nondiseased cells.

## Discussion

A wealth of evidence indicates that oxidative stress both enhances cellular aging and accelerates telomere dysfunction[12]. Here, we demonstrate a direct causal link between these two ROS-induced cellular outcomes. Oxidative stress was proposed to hasten telomere shortening and the onset of senescence by producing 8oxoG lesions in highly susceptible TTAGGG repeats[18]. Whether telomeric 8oxoG has a causal role in driving senescence could not be tested previously, because telomeres comprise a tiny fraction of the genome, and oxidants used to produce 8oxoG modify numerous cellular components and alter redox signaling. We overcame these barriers by using a precision chemoptogenetic tool that induces ${}^1O_2$-mediated 8oxoG formation exclusively at telomeres. We demonstrate that acute telomeric 8oxoG formation at telomeres is sufficient to trigger rapid premature senescence in the absence of telomere shortening or losses in primary and hTERT-expressing human cells. Instead, we observed telomere fragility, DDR signaling and replication stress at telomeres. Mechanistically, our data are consistent with a model (Fig. 7) in which 8oxoG itself, and/or repair intermediates, stall DNA replication at the telomeres, leading to a robust

induction of p53 signaling to arrest cell growth and enforce premature senescence.

We found that 8oxoG formation exclusively at telomeres induces multiple hallmarks of premature senescence, including increased SA-β-gal activity, nuclear area, CCFs, SASP and mitochondrial activity, and reduced cell growth, colony formation, Lamin B1 expression, EdU incorporation and RB phosphorylation. These phenotypes arise in replicative senescence, or oncogene and DNA-damaged-induced premature senescence; however, their rapid onset by a small base modification at the telomeres was surprising[5]. Several of these phenotypes were rescued by pharmacologic ATM inhibition or genetic p53 deletion, consistent with other models of premature senescence, and confirming DDR signaling causality[62,63]. The rapid timescale of telomeric 8oxoG-induced senescence would not typically allow for extensive telomere shortening, in agreement with our results. Notably, HeLa FAP-TRF1 cells displayed telomere shortening and losses only after chronic 8oxoG formation, especially in OGG1ko cells[28], raising the possibility that chronic damage may also accelerate shortening in nondiseased cells. Nevertheless, our data demonstrate that telomeres are profoundly sensitive to oxidative stress-induced 8oxoG. We propose the sensitivity results from DNA replication slowing or stalling, resulting in a robust DDR that is independent of shortening.

Telomeres exist in a 't-loop' structure organized by shelterin to prevent erroneous recognition as DSBs. Shelterin proteins can directly prevent HR and end-joining pathways from acting at telomeres even when the t-loop is absent but the DDR is activated (intermediate state), thereby preventing fusions[51,64]. These observations, together with our results, highlight how readily damaged telomeres can be sensed by the DDR and suggest that 8oxoG may promote an intermediate state. Telomeric 8oxoG did not disrupt shelterin localization to telomeres, consistent with our observation of telomere DDR in the absence of chromosome fusions and bridges or telomere shortening and losses. In yeast, loss of t-loops occurs with replicative aging, suggesting that impaired telomere organization may be a conserved feature of senescence[65].

Our data suggest that 8oxoG disrupts DNA replication at telomeres. Replication stress is defined as the slowing or stalling of DNA replication forks, and robustly increases telomere fragility[54,57]. While structurally undefined, fragile telomeres are believed to represent unreplicated regions in the telomere causing altered chromatinization[66]. A single induction of telomeric 8oxoG enhanced telomere fragility and activated ATR/Chk1. Replication stress also leads to under-replicated DNA, which can be repaired by MiDAS. Telomeric 8oxoG induced a robust increase in single chromatid MiDAS events, which is consistent with other models of telomere replication stress[59,60]. Increases in both telomere fragility and telomere MiDAS occur in cells depleted for TRF1, POT1, BRCA2 and RAD51, or when stressed by aphidicolin, oncogene overexpression or ATRi. In contrast, fragility and MiDAS decrease in cells depleted for downstream factors, including SLX4 and POLD3, and MiDAS is RAD52-dependent[58,60,67–70]. While the connection between MiDAS and telomere fragility is unclear, both phenotypes are increased in cells experiencing general or telomere specific replication stress, consistent with our results. Supporting a role for replication in 8oxoG-induced senescence, 8oxoG generated a much more robust telomere DDR in replicating cells compared with quiescent cells. Specifically, 8oxoG failed to significantly increase the percentage of quiescent cells showing four or more DDR+ telomeres—a phenotype previously correlated with replicative senescence[50]. Quiescence rescued the senescence phenotypes, demonstrating that telomere 8oxoG-induced senescence is due to replication stress.

How does 8oxoG impact replication? While studies have focused largely on the mutagenic consequences of 8oxoG, we argue that mutagenesis is unlikely to be a primary driver of the senescence phenotypes, since DDR foci arose immediately after lesion induction. Our data suggest that 8oxoG stalls replication at the telomeres.

8oxoG is a weak impediment to replicative DNA polymerase delta (Pol δ) in vitro, compared with bulky lesions from UV light or cisplatin. However, Pol δ stalls at 8oxoG, especially when incorporating C, even in the presence of its accessory factors[71,72]. Further support for Pol δ stalling derives from evidence that translesion polymerases η and λ function in 8oxoG bypass[72,73]. Moreover, most polymerase reactions were conducted using dNTP concentrations above relevant cellular concentrations, on nontelomeric templates[71]. The human mitochondrial replisome stalls substantially at 8oxoG in reactions containing cellular dNTP levels[74], suggesting that previous biochemical studies may have underestimated the impact of 8oxoG on replication fork progression in cells. Since difficult-to-replicate sequences, such as telomeres, themselves can impede Pol δ upon replication stress[75], future biochemical studies are warranted to study Pol δ synthesis using physiological dNTP levels and 8oxoG within telomere templates.

The key finding from our study that a small, nondistorting oxidative base lesion within telomeres is sufficient to induce premature senescence in the absence of telomere shortening is surprising, but provides a mechanistic explanation for telomere dysfunction foci arising in vivo in various contexts[27]. Mouse cardiomyocytes and baboon hepatocytes in vivo, show increased DDR+ telomeres with age, with no appreciable shortening despite the presence of senescence markers[20,76]. Oxidative stress is implicated in generating DDR+ telomeres in liver and intestinal cells, also without shortening, in mouse models of liver damage and chronic low-grade inflammation, respectively[19,24]. Human melanocytic nevi senesce in the absence of telomere shortening[77]. Moreover, in cell culture models of replicative senescence and ionizing radiation or $H_2O_2$-induced premature senescence, DDR foci persist or accumulate at telomeres, long after disappearing from nontelomere sites, irrespective of telomere length[50,76,78]. Together, these reports demonstrate that cells can senesce independent of telomere attrition under oxidative stress, and show DDR+ telomeres. Our finding that 8oxoG does not induce telomere shortening in an acute treatment, but significantly elevates DDR signaling, provides a possible mechanism. Consistent with previous work[50], we also observed the vast majority of γH2AX foci at chromatid ends were positive for telomere staining, indicating the DDR was not due to telomere loss. We propose that because telomeres are exquisitely sensitive to oxidative stress, they act as tumor suppressors even before they become critically short, and enforce senescence to prevent cellular transformation.

Importantly, our study demonstrates premature senescence in primary and nondiseased human cells following induction of a common, physiological oxidative DNA lesion targeted to the telomere. Oxidative stress is a ubiquitous source of DNA damage that humans experience due to endogenous metabolism and inflammation, exogenous environmental sources as well as life-stress, and 8oxoG levels are elevated in aged humans[79,80]. Our results highlight the importance of understanding how and where this DNA lesion arises within human genomes, since its presence at telomeres alone is sufficient to rapidly advance cellular aging. While other oxidative lesions may also contribute to telomere instability, 8oxoG is among the most abundant. In summary, our studies reveal a new mechanism of telomere-driven senescence linked to oxidative stress.

## Online content

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

## Methods

**Cell culture and cell line generation.** hTERT-expressing BJ and RPE1 cells, as well as primary BJ cells were purchased from ATCC and tested for mycoplasma. BJ cells were grown in DMEM (Gibco) with 10% Hyclone FBS and 1% penicillin/streptomycin. RPE cells were grown in DMEM/F12 (Gibco) with 10% FBS (Gibco) and 1% penicillin/streptomycin. To generate FAP-mCER-TRF1 expressing clones, HEK 293T cells were transfected with pLVX-FAP-TRF1 and Mission Packaging Mix (Sigma) to produce lentivirus. hTERT BJ and RPE1 cells were infected with virus 48 and 72 h post-transfection and then selected with 1 mg ml$^{-1}$ G418 (Gibco). Surviving cells were single-cell cloned and expanded before checking for FAP-mCER-TRF1 expression, and then referred to as BJ and RPE FAP-TRF1 cells. Primary BJ cells were infected and selected the same way, but were not single-cell cloned. After initial selection, FAP-TRF1 expression was maintained with 500 μg ml$^{-1}$ G418. U2OS and HeLa FAP-TRF1 cells were described previously[28]. Except for 293T cells, all cells were cultured at 5% O$_2$.

To generate ko cell lines, 293T cells were transfected with pLentiCRISPR V2 plasmids encoding guide RNAs to the respective targets and *Streptococcus pyogenes* Cas9 (GeneScript). FAP-TRF1 expressing cells were infected with lentivirus as above and selected with 1 μg ml$^{-1}$ (BJ) or 15 μg ml$^{-1}$ (RPE) Puromycin (Gibco). After selection and death of uninfected cells, the infected cells were expanded and expression of targeted protein(s) was determined by western blotting.

**Cell treatments.** For DL treatments, cells were plated at an appropriate density for the experiment overnight. The next day, cells were changed to Optimem (Gibco) and incubated at 37 °C for 15 min before adding 100 nM MG2I for another 15 min. Cells were then placed in the lightbox and exposed to a high intensity 660 nm LED light at 100 mW cm$^{-2}$ for 5 min (unless indicated otherwise). KBrO$_3$ and ETP were added in Optimem at the indicated concentrations for 1 h.

**Growth analyses.** For cell counting experiments, cells were plated at a low density in six-well or 6 cm plates overnight. Cells were treated as indicated and returned to the incubator and recovered for the indicated amount of time (typically 4 days). Cells were detached from the plates, resuspended and counted on a Beckman Coulter Counter. Each experiment had two to three technical replicates, which were averaged.

**Senescence-associated β-gal assay.** We detected β-gal activity according to the manufacturer's instructions (Cell Signaling). Briefly, cells were washed with PBS, and then fixed at room temperature for 10 min. After two rinses with PBS, cells were incubated overnight at 37 °C with X-gal staining solution with no CO$_2$. Images were acquired with a Nikon brightfield microscope with a DS-Fi3 camera. Images were scored in NIS-Elements (Nikon). At least 300–800 cells were counted per condition for each experiment.

**Colony formation assay.** RPE FAP-TRF1 cells were plated in 6 cm plates overnight. The cells were treated with DL the next day and immediately detached, counted and plated in triplicate in six-well plates. After 7–8 days, the colonies were fixed on ice in 100% methanol, stained with crystal violet solution and then counted manually.

**Immunofluorescence and fluorescence in situ hybridization.** Cells were seeded on coverslips and treated as indicated. Following treatment and/or recovery, cells were washed with PBS and fixed at room temperature with 4% formaldehyde. If cells were extracted before fixation, they were treated on ice with ice-cold CSK buffer (100 mM NaCl, 3 mM MgCl$_2$, 300 mM glucose, 10 mM Pipes pH 6.8, 0.5% Triton X-100 and protease inhibitors tablet (Roche)). Fixed cells were rinsed with 1% BSA in PBS, and washed three times with PBS-Triton 0.2% before blocking with 10% normal goat serum, 1% BSA and 0.1% Triton X. Cells were incubated overnight at 4 °C with indicated primary antibodies. The next day, cells were washed three times with PBS-T before incubating with secondary antibodies and washing again three times with PBS-T. If fluorescence in situ hybridization (FISH) was performed, the cells were refixed with 4% formaldehyde, rinsed with 1% BSA in PBS and then dehydrated with 70%, 90% and 100% ethanol for 5 min. Telomeric PNA probe was diluted 1:100 (PNABio) prepared in 70% formamide, 10 mM Tris-HCl pH 7.5, 1× Maleic Acid buffer, 1× MgCl$_2$ buffer and boiled for 5 min before returning to ice. Coverslips were then hybridized in humid chambers at room temperature for 2 h or overnight at 4 °C. The cells were washed twice with 70% formamide and 10 mM Tris-HCl pH 7.5, three times with PBS-T then rinsed in water before staining with 4′,6-diamidino-2-phenylindole (DAPI) and mounting. Image acquisition was performed with a Nikon Ti inverted fluorescence microscope. Z-stacks of 0.2 μm (×60 objective) or 0.5 μm (×20 objective) thickness were captured and images were deconvolved using the NIS-Elements Advance Research software algorithm. For MN analysis, at least 30 MN were analyzed per experiment.

To detect EdU incorporation, Click chemistry was performed after the secondary antibody washes according to the manufacturer's instructions (Thermo).

**Live-cell imaging.** BJ FAP-TRF1 cells were infected with H2B-mCherry (pCSII-EF) and then sorted on a MoFlo Astrios for mCherry positive cells. For imaging, cells were plated on poly-D-lysine (0.5 mg ml$^{-1}$) treated glass-bottomed plates (Cellvis P06-1.5H-N). After treatment, cells were imaged at ×20 on a Nikon TiE with a humidified chamber at 37 °C every 4 min for mCherry signal and DIC. Each well was imaged 16 times (4 × 4) and image registration was used to stich the images together.

Mitotic events were scored manually. Each dividing cell was tracked for the duration of the time-lapse; if a MN arose following mitosis and persisted for more than two frames, it was scored as MN+.

**Metaphase spreads.** Chromosome spreads were prepared by incubating cells with 0.05 μg ml$^{-1}$ colcemid for 2 h before harvesting with trypsin. Cells were incubated with 75 mM KCl for 8 min at 37 °C and fixed in methanol and glacial acetic acid (3:1). Cells were dropped onto washed slides and dried overnight before fixation in 4% formaldehyde. Slides were treated with RNaseA and Pepsin at 37 °C, and then dehydrated. FISH was performed as above, and included a CENPB (PNABio) probe in addition to the telomere probe. Numbers are normalized to 46 chromosomes per cell.

**Pulsed-field gel electrophoresis of cells in agarose plugs.** Double-stranded DNA breaks (DSBs) were detected as previously described. Briefly, cells were harvested by trypsinization, washed with PBS and counted. A total of 500,000 cells were embedded in 0.75% Clean Cut Agarose and allowed to solidify before digesting overnight with Proteinase K at 50 °C. The plugs were washed four times for 1 h before loading onto a 1% agarose gel. The gel was run with 0.5× TBE at 14 °C with a two-block program; block 1: 12 h, 0.1 s initial, 30 s final, at 6 V cm$^{-1}$; block 2: 12 h 0.1 s initial, 5 s final, 3.8 V cm$^{-1}$. The gel was then dried 2 h at 50 °C before staining with SYBR Green and imaging on a Typhoon.

**XRCC1 recruitment and analysis.** RPE FAP-TRF1 cells were plated on coverslips so they would be around 70% confluent the next day. They were then transfected with pEYFP-XRCC1 (1 μg) and 6 μl Fugene 6 (Promega) in Optimem (Gibco) using medium without antibiotics. After 24 h, the cells were treated with DL for 10 min and then immediately subjected to CSK extraction before fixation. After washing, cells were mounted without DAPI. Only YFP-positive cells were imaged and the CFP channel was used to mark telomeres (FAP-mCER-TRF1).

**Detection of 8oxoG in telomere DNA.** After treatment, cells were immediately scraped on ice and DNA was isolated with antioxidants 100 mM of butylated hydroxytoluene (Sigma; DMSO solvent) and deferoxamine mesylate (Sigma; water solvent) as previously described[28]. DNA was treated with FPG (NEB, 1.3 U μg DNA$^{-1}$) and then digested with *Rsa*I and *Hinf*I overnight. FPG sensitive sites were converted to DSBs with 2 U S1 nuclease treatment for 15 min at 37 °C, before running pulsed-field gel electrophoresis (PFGE) and Southern blotting as previously described[28].

**Image acquisition and analysis.** All immunofluorescence (IF) images were acquired on a Nikon Ti inverted fluorescent microscope equipped with an Orca Fusion cMOS camera or CoolSNAP HQ2 CCD. Z-stacks were acquired for each image and deconvolved using blind, iterative methods with NIS-Elements AR software.

For colocalizations, deconvolved images were converted to Max-IPs and converted to a new document. The object counts feature in NIS AR was used to set a threshold for foci that was kept throughout the experiment. The binary function was used to determine the intersections of two or three channels in defined regions of interest (ROI) (DAPI-stained nuclei). For whole nuclei signal intensity, the automated measurements function was used on ROIs.

**Western blotting.** Cells were collected from plates with trypsin, washed and then lysed on ice with RIPA buffer (Santa Cruz) supplemented with PMSF (1 nM), 1× Roche Protease and Phosphatase Inhibitors and Benzonase (Sigma catalog no. E8263; 1:500) for 15 min and then incubated at 37 °C for 10 min, before centrifuging at 14,000g for 15 min at 4 °C. Protein concentrations were determined with the BCA assay (Pierce) and 10–30 μg of protein was electrophoresed on 4–12% (or 12% for OGG1ko blot) Bis-Tris gels (Thermo) before transferring to polyvinylidene difluoride membranes (GE Healthcare). Membranes were blocked in 5% milk and blotted with primary and secondary horseradish peroxidase antibodies. Signal was detected by enhanced chemiluminescence detection and X-ray film.

**Reverse transcription qPCR.** RNA was extracted from cells using the Qiagen RNeasy Plus Mini kit; 500–1,000 ng RNA was converted to cDNA using the High capacity RNA-to-cDNA Kit (Thermo). cDNA (50 ng) was subjected to real-time qPCR using Taqman probes at 1× and the Taqman Universal PCR kit (Thermo). Data were analyzed using the delta delta Ct method.

**Flow cytometry.** To analyze apoptosis, cells were treated as indicated and allowed to recover for 4 days. Floating cells were collected, and then attached cells were collected with trypsin and combined. After centrifugation and washing, the cells were incubated with Alexa Flour 488 annexin V and 1 μg ml$^{-1}$ PI in 1× annexin-binding buffer for 15 min in the dark (Thermo). After resuspending in additional binding buffer, the cells were analyzed on an Accuri C6 (Beckman) using FL1 and FL3.

For cell cycle analysis, 23 h after treatment, cells were pulsed with 20 μM EdU and incubated for an additional hour (Thermo). Cells were collected with trypsin, washed with 1% BSA in PBS and then fixed with Click-IT fixative D. After washing with 1% BSA in PBS, the cells were permeabilized with 1× component E for 15 min, before performing Click chemistry with Alexa Flour 488 azide for 30 min in the dark. Cells were washed with 1× component E, and then resuspended in 500 μl FxCycle PI/RNase (Thermo) for 15 min before analyzing on Accuri C6. Standard gating for cells versus debris and singlet was conducted.

**Seahorse analysis.** OCR was measured using a SeahorseXF96 Extracellular Flux Analyzer (Seahorse Bioscience) essentially as previously described[81]. After treatment and recovery for the indicated times, cells were seeded in XF96 cell culture plates at $8 \times 10^4$ cells per well in the presence of Cell-Tak cell and tissue adhesive. Cells were then washed and growth medium was replaced with bicarbonate-free medium. Thereafter, cells were incubated for another 60 min in a 37 °C incubator without $CO_2$ followed by simultaneous OCR measurements.

**Analysis of secreted proteins.** Cells were treated as indicated, and recovered for 7 days. Medium was collected and debris pelleted by centrifugation for 10 min. Media were stored at −80 °C until ready for analysis. The indicated analytes were assessed for concentration with multiplex ELISA (Luminex). Each sample was analyzed in duplicate, and a blank medium sample was analyzed for background levels. After determining concentrations alongside standard curves, the values were adjusted for the number of cells present at the time of harvest.

**Bulk RNA-seq.** RNA was prepared using the Qiagen RNeasy Mini Plus kit; 1 μg total RNA was sent to Genewiz for library preparation and sequencing. RNA with a RNA integrity number >9 was polyA selected and fragmented before cDNA synthesis. Adapters were ligated, PCR enriched and then sequenced on a HiSeq 2×150 in paired-end mode. Each sample was sequenced to at least 30 million reads.

The triplicate measurements of gene expression in the mRNAseq data was quantified using Salmon (v.0.7.2) to the HG19 refseq transcript annotations[82]. Unique genes were obtained by summing across transcript isoforms and gene count matrixes from untreated and treated ('DL') and were analyzed with DEseq2 to obtain fold change and $P$ value scores for each gene[83]. Differentially expressed genes were defined as $>0.5 \log_2 FC$, $-\log_{10}(P) > 10^4$ and quantifiable ('expressed', >10 counts) in all three cell lines. To determine if gene expression was altered in chromosomal regions near the telomeres (which may be exposed to oxidative stress from the FAP system), we binned genes by the distance of their start site from the chromosomal ends and averaged across genes of a given distance from the chromosome end. We performed gene set enrichment using FGSEA and Hallmark gene sets[84].

**Telomere restriction fragment analysis.** Telomere restriction fragment (TRF) analysis was performed as previously described[28]. Briefly, genomic DNA was extracted from cells using Qiagen Tip-20 or 100 according to the manufacturer's instructions. The DNA was digested with *Hinf*I and *Rsa*I overnight, before PFGE. After drying the gel, the molecular weight ladder was detected with SYBR Green (Thermo) and then hybridization with a $^{32}$P-labeled telomere probe was carried out as described.

**Telomere shortest length assay.** The TeSLA assay was performed as previously described with some modifications[85]. Genomic DNA (50 ng) was ligated to TeSLA-T oligo before digestion with *Cvi*AII to create 5′ AT overhangs. DNA was further digested with *Bfa*I, *Nde*I and *Mse*I (NEB) to generate 5′ TA overhangs. After dephosphorylation, AT and TA adapters were ligated, and then PCR (four per sample) was performed with AP and TeSLA-TP primers. PCR product was cleaned using a Genejet PCR Purification kit before electrophoresis. Telomere fragments were detected after drying the gel using in-gel hybridization as previously described[28].

**Metaphase IF.** For metaphase IF (Meta-TIF), cells were collected by trypsinization, washed with PBS, counted and centrifuged. Then, 200,000 cells were swelled in 0.2% potassium chloride and sodium citrate for 5 min at 37 °C and cytocentrifuged onto slides (10 min, 2,000 r.p.m., medium acceleration). Cells were fixed 4% formaldehyde in PBS then processed for IF/FISH as described above.

**Detection of mitotic DNA synthesis.** At 16 h after treatment, cells were incubated with 7 μM Cdk1 inhibitor RO3306 (Millipore) for 24 h. Cells were washed with PBS, and then released into medium with 20 μM EdU and colcemid for 1 h before harvesting by mitotic shake-off. Metaphase spreads were prepared as described above and EdU staining performed using a Click-iT EdU Alexa Fluor 594 imaging kit (ThermoFisher) after FISH staining.

**Antibodies.** GFP (Abcam catalog no. ab6556), TRF1(Abcam catalog no. ab10579), GAPDH (Santa Cruz catalog no. sc-47724), OGG1 (Abcam catalog no. ab124741), Actin Cell Signaling catalog no. 3700), Lamin B1 Abcam catalog no. ab16048), Lamin A/C (Cell Signaling catalog no. 4777), γH2AX (Santa Cruz catalog no. sc-517348), 53BP1 (Novous catalog no. NB100-304), TRF2 (Novous catalog no. NB110-57130), MDM2 (Cell Signaling catalog no. 86934), p53(Santa Cruz catalog

no. sc-126), p21 (Cell Signaling catalog no. 2947), p16 (Proteintech catalog no. 10883-1-AP), pRB S807/811 (Cell Signaling catalog no. 8516), pCHK2 T68(Cell Signaling catalog no. 2197), pCHK1 S317 (Cell Signaling catalog no. 12302), pATM S 1981 (Abcam catalog no. ab81292), CHK1 (Cell Signaling catalog no. 2360), H3K27me3 (Cell Signaling catalog no. 9733), H3K27Ac (Cell Signaling catalog no. 8173), LSD1 (Cell Signaling catalog no. 2184), cGAS (Cell Signaling catalog no. 66546), p62 (Cell Signaling catalog no. 39749).

**Chemical reagents.** Potassium bromate KBrO3 (Sigma catalog no. 309087; CAS: 7758-01-2), sodium azide NaN3 (Fisher Chemical catalog no. S227I; CAS: 26628-22-8), ATMi KU60019 (Selleckchem catalog no. S1570), Cdk1 inhibitor IV, RO-3306 (Millipore catalog no. 217699), ETP (Cell Signaling catalog no. 2200; CAS 33419-42-0), Aphidicolin (Santa Cruz catalog no. sc-201535; CAS 38966-21-1).

**Recombinant DNA.** pLentiCRISPR v2 gRNA, OGG1 targeting sequence (exon 4:GCTACGAGAGTCCTCATATG), pLentiCRISPR v2 gRNA p53 targeting sequence (exon 3:CCCCGGACGATATTGAACAA), pLentiCRISPR v2 gRNA CDK2NA targeting sequence (exon 3:GGCCTCCGACCGTAACTATT), (GeneScript). pEYFP-XRCC1 plasmid (gift from M. Otterlei (NTNU, Norway)).

**Statistics and reproducibility.** The number of biological and technical replicates are noted in figure legends and Methods. Except for the RNA-seq data, all statistical analysis was done in Graphpad Prism 9. No statistical method was used to predetermine sample size. Rare outliers indicated as blue, determined by Graphpad, in the source data for Fig. 6b,c and Extended Data Fig. 2a were omitted. Investigators were not blinded to allocation during experiments and outcome assessments.

**Contact for reagent and resource sharing.** Further information and requests for reagents should be directed to and will be fulfilled by the lead contact, P.L.O. (plo4@pitt.edu).

**Reporting summary.** Further information on research design is available in the Nature Research Reporting Summary linked to this article.

## Data availability
All relevant data are available in the Source Data provided with this paper or from the authors upon reasonable request. The mRNAseq dataset are deposited at GEO (GSE175686).

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

## Acknowledgements
This work was supported by National Institute of Health (NIH) grants F32AG067710-01 and K99ES033771 (to R.P.B.), R35ES030396 and R01CA207342 (to P.L.O.), and R01EB017268 (to M.P.B.) and R35ES031638 (to B.V.H.). This work was supported by the Glenn Award for Research in Biological Mechanisms of Aging (to P.L.O.). This project was also supported by the UPMC Hillman Cancer Center Postdoctoral Fellowship for Innovative Cancer Research (R.P.B.). We thank R. O'Sullivan, E. Fouquerel, A. Gurkar and K. Aird for careful reading of the manuscript. We also thank S. Sanford for help with artwork. This project used the UPMC Hillman Cancer Center Cytometry Facility and Luminex Core Laboratory at the Cancer Proteomics Facility that are supported in part by award P30CA047904.

## Author contributions
R.P.B. and P.L.O. conceived the study. R.P.B. and P.L.O. designed the experiments. R.P.B. performed most of the experiments. M.d.R. conducted RPE metaphase spread analysis and assisted with some β-gal and telomere DDR experiments. S.A.T. conducted qPCR experiments. A.C.D. performed TeSLA experiments and assisted with 8oxoG detection assays. V.R. and B.V.H. performed Seahorse analysis. J.S.-O. conducted the RNA-seq analyses. M.P.B. provided the MG2I dye and 660 nm LED irradiator. R.P.B. and P.L.O. wrote the manuscript with assistance from the other authors.

## Competing interests
M.P.B. is a founder in Sharp Edge Labs, a company applying the FAP-fluorogen technology commercially. The remaining authors declare no competing interests.

## Additional information

**Extended data** is available for this paper at https://doi.org/10.1038/s41594-022-00790-y.

**Correspondence and requests for materials** should be addressed to Patricia L. Opresko.

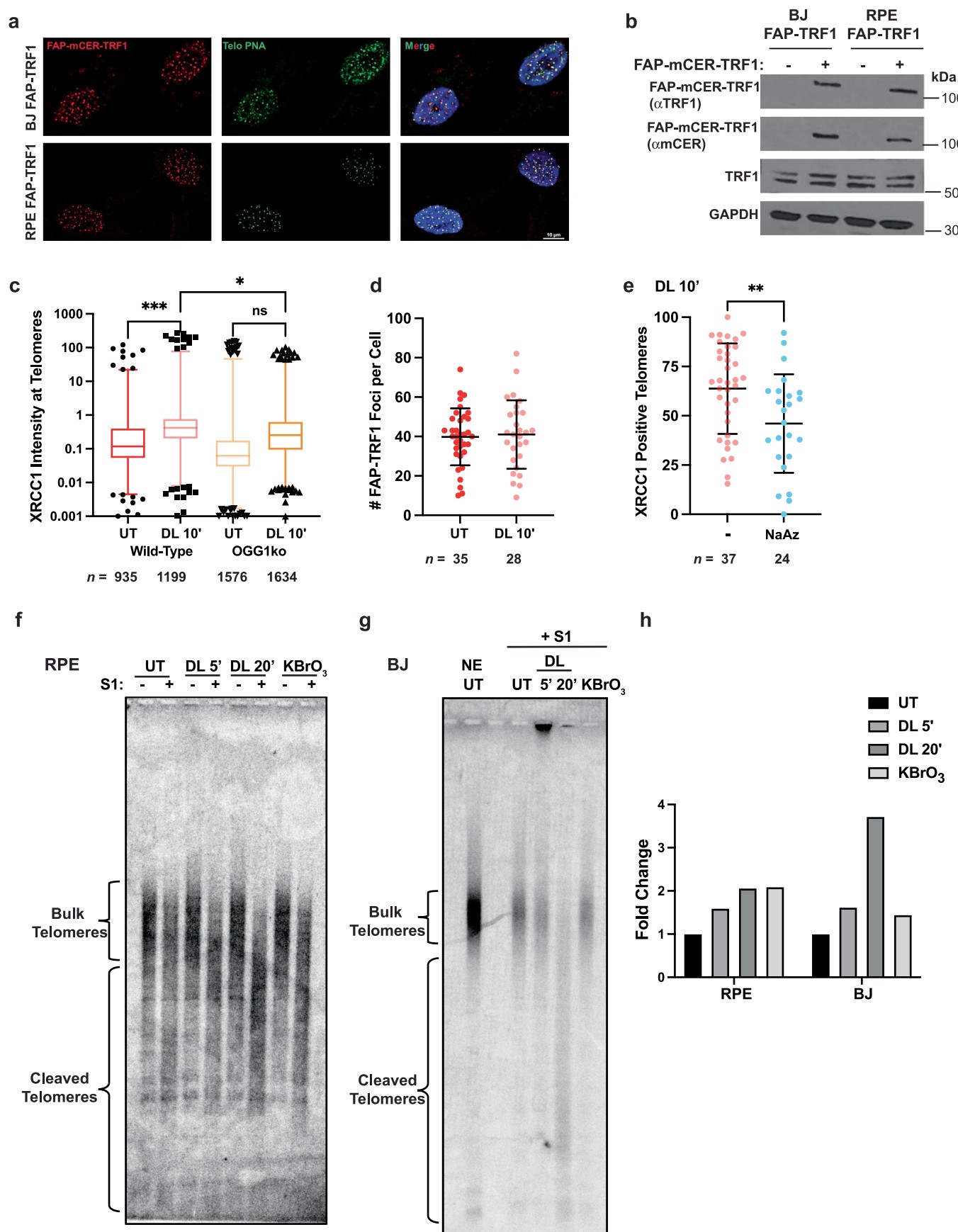

**Extended Data Fig. 1 | See next page for caption.**

**Extended Data Fig. 1 | Confirmation of telomeric 8oxoG induction in FAP-TRF1 expressing cells.** (**a**) Representative images of FAP-mCER-TRF1 colocalization with telomeres in BJ (top panel) and RPE (bottom) clones expressing FAP-TRF1 visualized by anti-mCER staining (red) and with telo-FISH (green). Scale bar = 10 μm. (**b**) Immunoblot for TRF1 in whole cell extracts from hTERT BJ and RPE cells with and without FAP-mCER-TRF1 expression. TRF1 antibody detects both exogenous and endogenous TRF1 while mCER antibody detects exogenous only. (**c**) Quantification of YFP-XRCC1 signal intensity at telomeric foci as shown in (Fig. 1a) normalized to CFP signal. Box plots represent median and 25th to 75th percentiles, and whiskers represent the $1^{st}$ to 99th percentiles. Data derived from the indicated $n$ number of foci analyzed. Statistical analysis was by one-way ANOVA (ns = not significant, * $p < 0.05$, *** $p < 0.001$). (**d**) Quantification of number (#) of FAP-mCer-TRF1 foci per cell by direct mCer visualization of untreated cells (UT) or 10 min after dye + light (DL 10'). Error bars represent the mean ± s.d. from the indicated $n$ number of nuclei analyzed. Statistical analysis by two-tailed t-test (p value was not significant). (**e**) Quantification of percent YFP-XRCC1 positive telomeres per nuclei after 10 min dye + light with pretreatment of 100 μM $NaN_3$ for 15 min prior to light exposure (NaAz) or with no pretreatment (-). Error bars represent the mean ± s.d. from the indicated $n$ number of nuclei analyzed. Statistical analysis by two-tailed t-test (** $p < 0.01$). (**f,g**) Detection of 8oxoG in telomeres. Genomic DNA isolated from RPE (**f**) and BJ (**g**) FAP-TRF1 cells following no treatment, 5 or 20 min dye + light or 40 mM $KBrO_3$, was treated with FPG glycosylase, and then treated with S1 nuclease (+) or not (-) as indicated. Intact and cleaved telomere restriction fragments were separated by PFGE, and telomeres were detected by Southern blotting. NE = no enzyme treatment for reference from UT cells. (**h**) The percent of cleaved telomeres was calculated and normalized to UT samples to quantify a fold change in telomere cleavage. For RPE the difference between -S1 and +S1 was used, and normalized to UT cells.

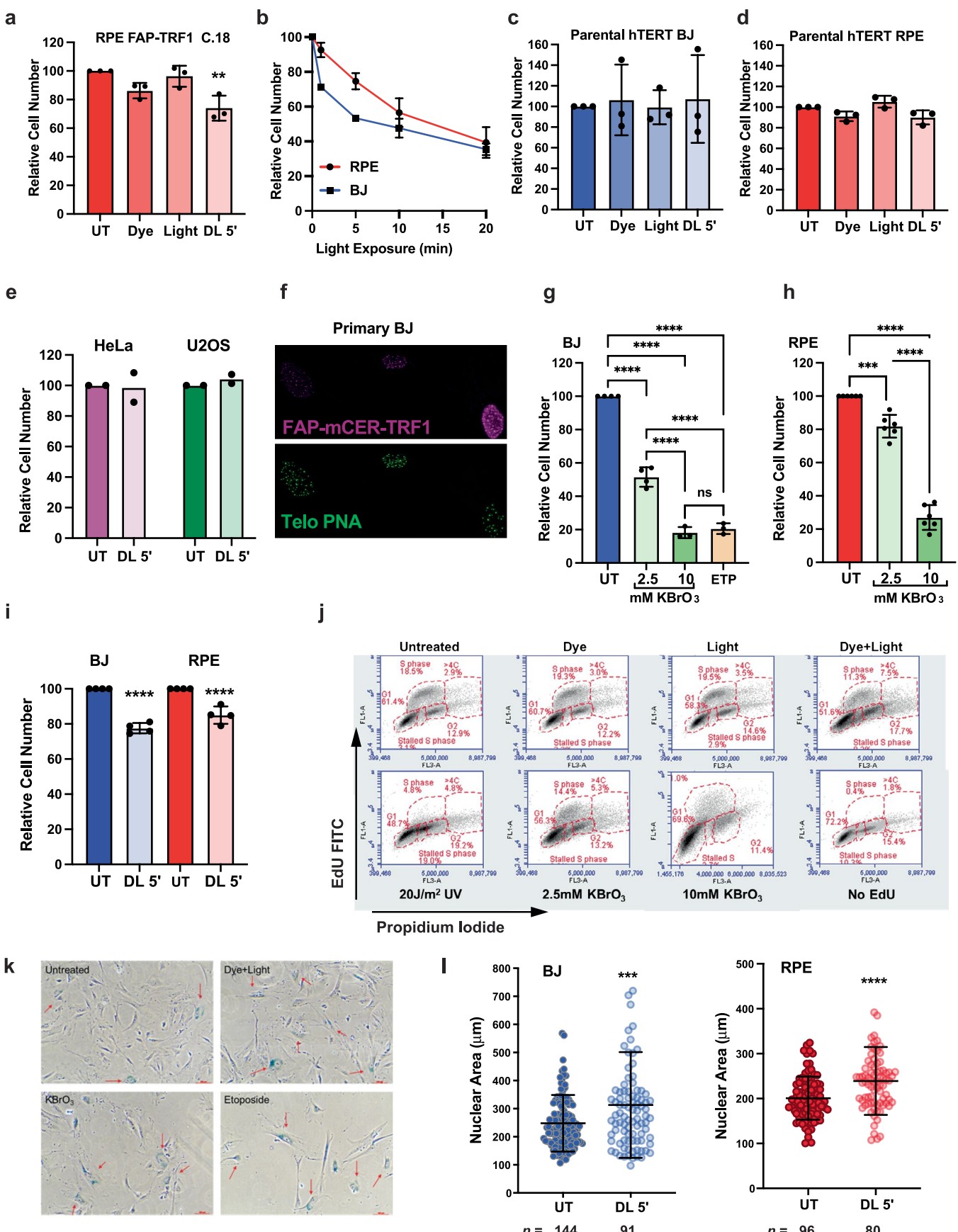

**Extended Data Fig. 2 | See next page for caption.**

**Extended Data Fig. 2 | Characterization of damage-induced growth reduction and senescent phenotypes.** (**a**) Cell counts of an additional RPE FAP-TRF1 clone (#18) obtained 4 days after recovery from indicated treatments. (**b**) Cell counts of RPE FAP-TRF1 (red) and BJ FAP-TRF1 (blue) cells 4 days after recovery from dye + light treatments relative to untreated. Error bars represent the means ± s.d. from three independent experiments. (**c-d**) Cell counts of parental BJ-hTERT (**c**) or RPE-hTERT (**d**) cells obtained 4 days after recovery from indicated treatments. (**e**) Cell counts of FAP-TRF1 expressing HeLa and U2OS clones obtained 4 days after recovery from 5 min dye + light treatment relative to untreated. Black circles indicate the number of independent experiments. (**f**) Representative image of FAP-mCER-TRF1 protein colocalization with telomeres in bulk population primary BJ cells visualized by mCER IF (pink) with telo-FISH (green). (**g-h**) Cell counts of BJ (**g**) or RPE (**h**) FAP-TRF1 cells 4 days after recovery from one hour treatments with 2.5 or 10 mM KBrO$_3$, and for BJ FAP-TRF1 with 50 μM etoposide (ETP). (**i**) Cell counts of BJ or RPE FAP-TRF1 cells obtained 24 hours after recovery from 5 min dye + light treatment relative to untreated. For panels **a, c-d** and **g-i**, error bars represent the mean ± s.d. from the number of independent experiments indicated by the black circles in the bar graphs. Statistical significance determined by one-way ANOVA (ns = not significant, **p < 0.01, ***p < 0.001, ****p < 0.0001). (**j**) Flow cytometry plots of RPE FAP-TRF1 cells showing gating based on EdU and propidium iodine staining for various cell cycle phases 24 h after no treatment or exposure to dye, light, 5' dye + light, 20 J/m² UVC, or one hour treatment with 2.5 or 10 mM KBrO$_3$. (**k**) Representative images of β-galactosidase positive BJ FAP-TRF1 cells obtained 4 days after recovery from indicated treatments (From Fig. 1h-i). Scale bar = 100 μm. (**l**) Size of nuclear area (μm²) of BJ (blue) or RPE (red) FAP-TRF1 cells obtained 4 days after recovery from no treatment or 5 min dye + light. Error bars represent the mean ± s.d. from the indicated *n* number of nuclei analyzed. Statistical analysis by two-tailed t-test (***p < 0.001, ****p < 0.0001).

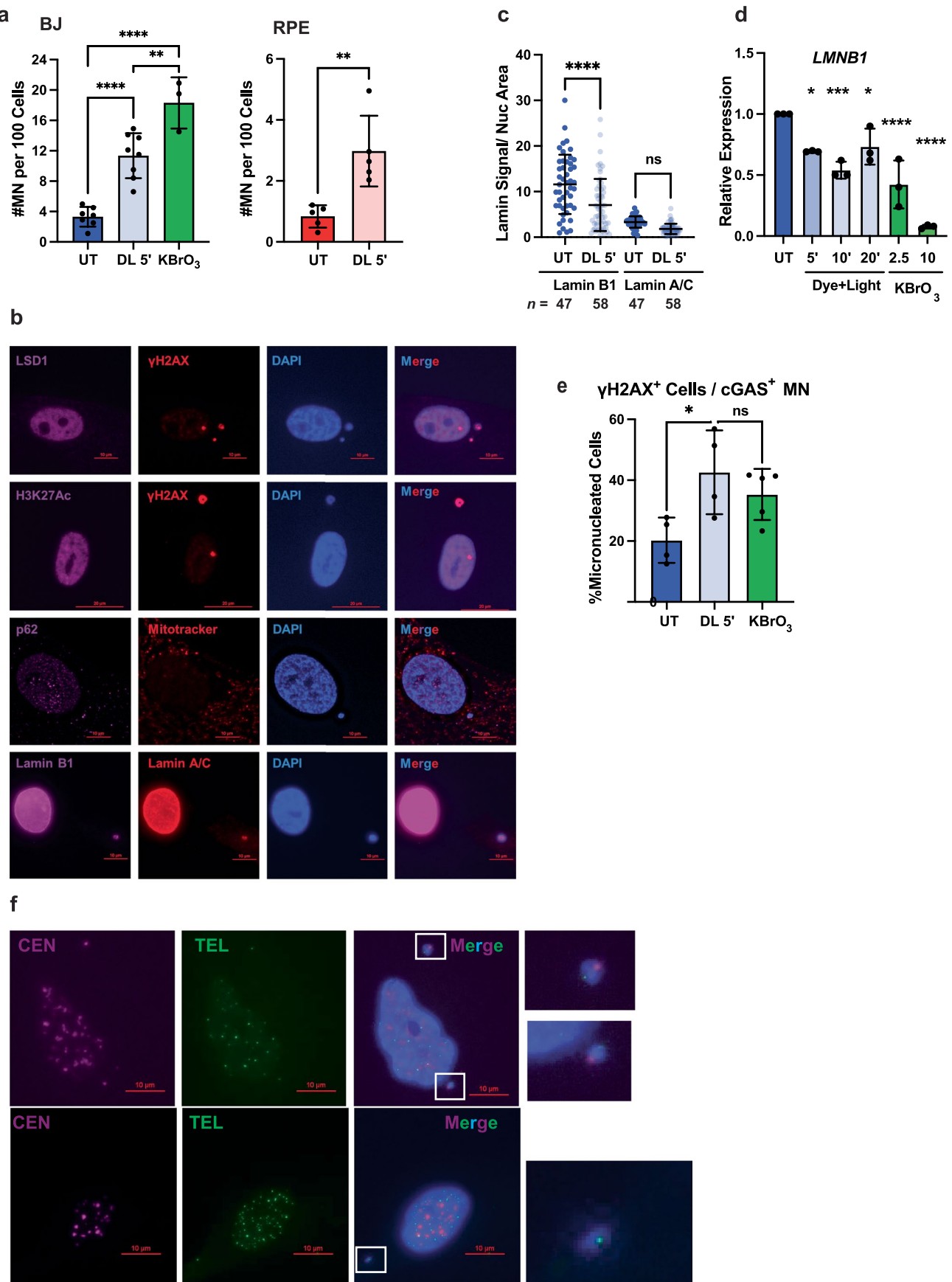

**Extended Data Fig. 3 | See next page for caption.**

**Extended Data Fig. 3 | Characterization of telomeric 8oxoG induced cytoplasmic DNA.** (**a**) Quantification of the number of MN per 100 nuclei for BJ and RPE FAP-TRF1 cell 4 days after 5 min dye + light, or for BJ after one hour 2.5 mM $KBrO_3$. At least 300 cells scored per experiment. (**b**) Representative images of RPE FAP-TRF1 cells stained for the indicated markers 4 days after 5 min dye + light treatment (related to Fig. 2b). Scale bar = 10 μm for rows 1, 3-4, and 20 μm for row 2. (**c**) Quantification of Lamin B1 and Lamin A/C signal intensity normalized to nuclear area from Panel (**b**). Error bars represent the mean ± s.d. from the indicated *n* number of nuclei analyzed. Statistical analysis by two-way ANOVA (ns = not significant, ****$p < 0.0001$). (**d**) Quantification of *LMNB1* mRNA from BJ FAP-TRF1 cells 4 days after 5, 10 or 20 min dye + light, or one hour 2.5 or 10 mM $KBrO_3$, relative to untreated. (**e**) Quantification of the percent of BJ FAP-TRF1 cells with micronuclei that show overall nuclear γH2AX staining and have a cGAS positive micronucleus 4 days after 5 min dye + light or one hour 2.5 mM $KBrO_3$ treatment. For panels **a** and **d-e**, error bars represent the mean ± s.d. from the number of independent experiments indicated by the black circles in the bar graphs. Statistical significance was determined by two-tailed t-test (panel **a** RPE) or one-way ANOVA (panel **a** BJ, and **d-e**). (ns = not significant, *$p < 0.05$, **$p < 0.01$, ***$p < 0.001$, ****$p < 0.0001$). (**f**) Representative images of RPE FAP-TRF1 cells 4 days after 5 min dye + light treatment and stained with centromeric and telomeric PNAs by FISH. White boxes zoom in on MN. Scale bar = 10 μm.

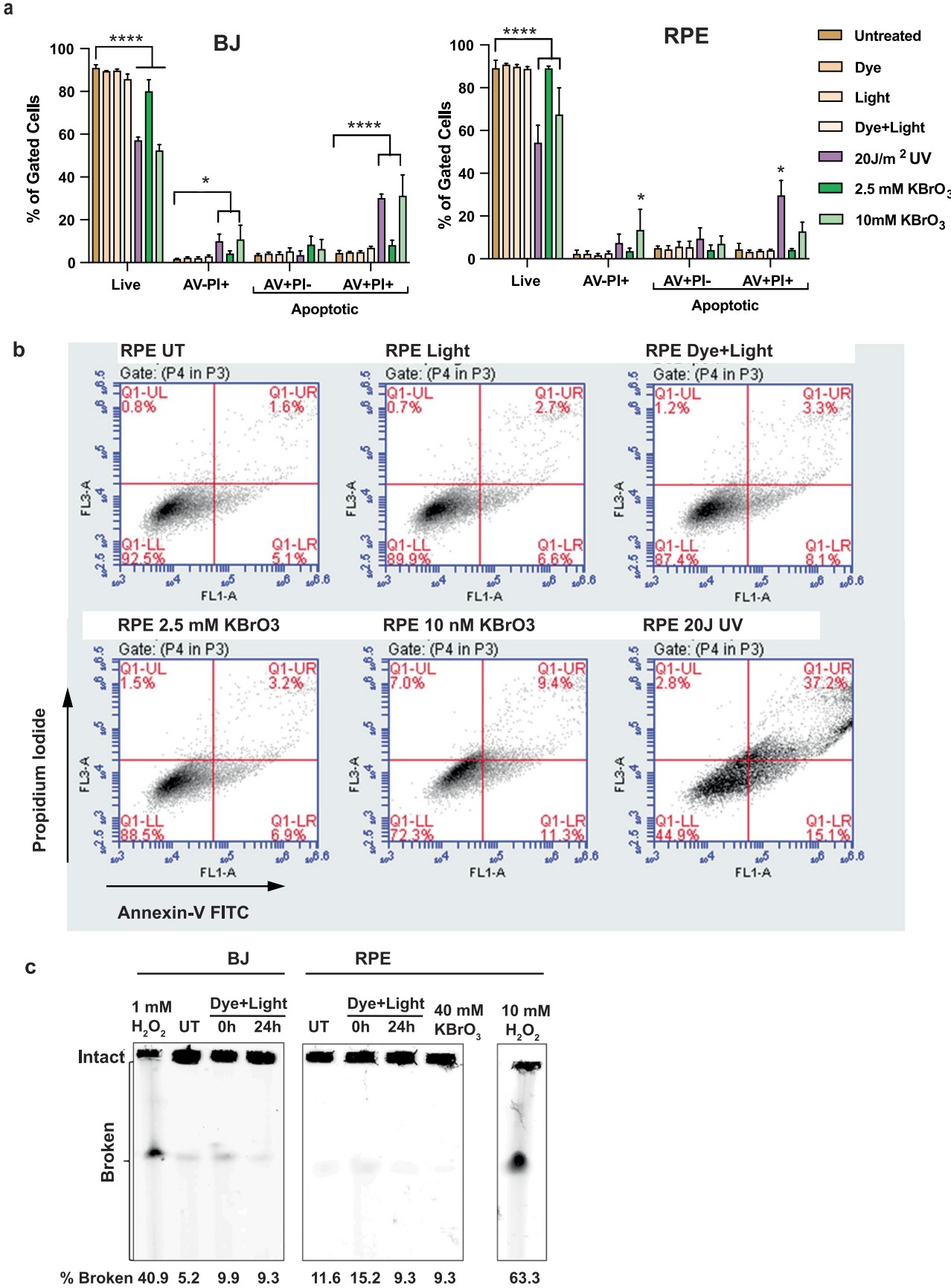

**Extended Data Fig. 4 | See next page for caption.**

**Extended Data Fig. 4 | Acute telomeric 8oxoG does not increase apoptosis or DNA double strand breaks.** (**a**) Percent of BJ or RPE FAP-TRF1 cells positive for annexin V (AV), propidium iodide (PI), or both 4 days after indicated treatments. Dye, light or dye + light was for 5 min. $KBrO_3$ exposure was for one hour. Error bars represent the mean ± s.d. from three independent experiments. Statistical analysis was by two-way ANOVA (*$p < 0.05$, ****$p < 0.0001$). (**b**) Representative scatterplots of Annexin V (y-axis) and propidium iodine (x-axis) staining of cells 4 days after the indicated treatments. (**c**) PFGE of cells in agarose plugs and SybrGreen staining of genomic DNA from BJ and RPE FAP-TRF1 cells untreated (UT) or after 0 or 24 h recovery from 5 min dye + light treatment. Treatments for one hour with 1 or 10 mM $H_2O_2$, or 40 mM $KBrO_3$, used as positive controls.

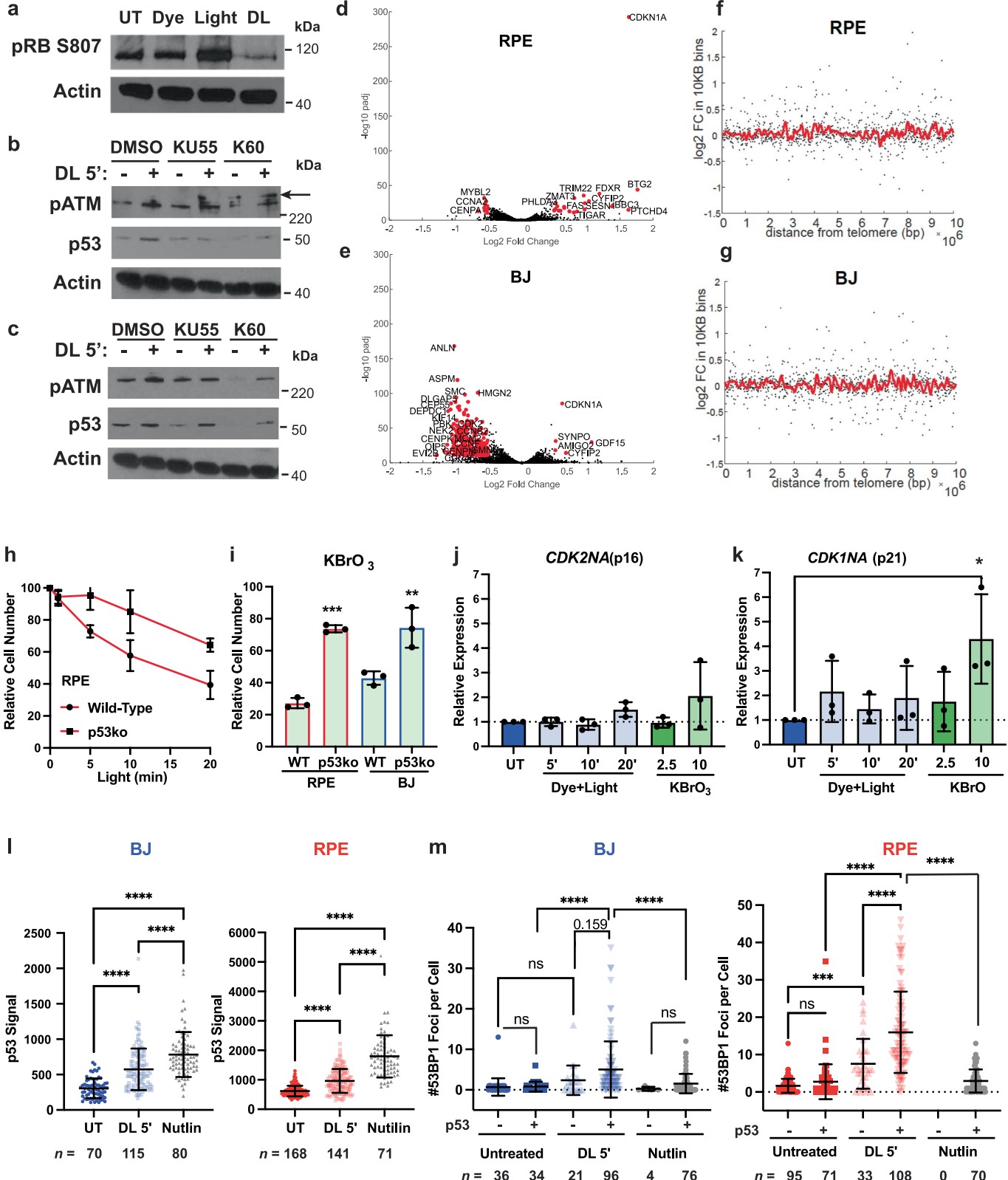

**Extended Data Fig. 5 |** See next page for caption.

**Extended Data Fig. 5 | 8oxoG induced DDR signaling and p53 activation. (a)** Immunoblot of phosphorylated RB from BJ FAP-TRF1 cells 4 days after dye, light or DL for 5 min. **(b, c)** Immunoblots of BJ (**b**) and RPE (**c**) FAP-TRF1 cells 3 hours after 5 min DL. Cells were pre- and post-treated with ATMi KU55933 (10 μM) or KU60019 (1 μM) or DMSO. Arrow indicates non-specific band. **(d, e)** Volcano plots of gene expression changes in RPE (**d**) and BJ (**e**) FAP-TRF1 cells 24 hours after dye + light. Each dot is a gene and red dots are significantly up or down-regulated. HeLa cells showed no significant changes. Analyzed with DEseq2. **(f, g)** Gene expression analysis in RPE (**f**) and BJ (**g**) FAP-TRF1 cells 24 hours after dye + light, as a function of chromosome position. Each dot is a 10 kb bin, and the red line = the average. **(h)** Counts of wild-type and p53ko RPE FAP-TRF1 cells 4 days after indicated treatment. Error bars represent the mean ± s.d. from three independent experiments. **(i)** Counts of wild-type and p53ko RPE and BJ FAP-TRF1 cells 4 days after treatment with 10 mM (RPE) or 2.5 mM (BJ) KBrO$_3$. **(j, k)** qPCR analysis of p16 mRNA (*CDK2NA*) and p21 mRNA (*CDK1NA*) in BJ FAP-TRF1 cells, 4 days after treatment. For panels **i-k**, error bars represent the mean ± s.d. from the number of independent experiments indicated by the black circles. Statistical analysis by one-way ANOVA (*$p < 0.05$, **$p < 0.01$, ***$p < 0.001$). **(l)** Quantification of p53 protein signal intensity by IF in BJ and RPE FAP-TRF1 cells 3 hours after 5 min DL or 20 μM nutlin. **(m)** 53BP1 foci per cell analyzed by IF from cells as treated in panel (**l**). p53 signal intensity by IF was used to determine p53 expression (+). For panel **l** and **m**, error bars represent the mean ± s.d. from the indicated *n* number of nuclei analyzed from two independent experiments. Statistical analysis by one-way ANOVA (**l**) or two-way ANOVA (**m**) (ns=not significant, ***$p < 0.001$, ****$p < 0.0001$).

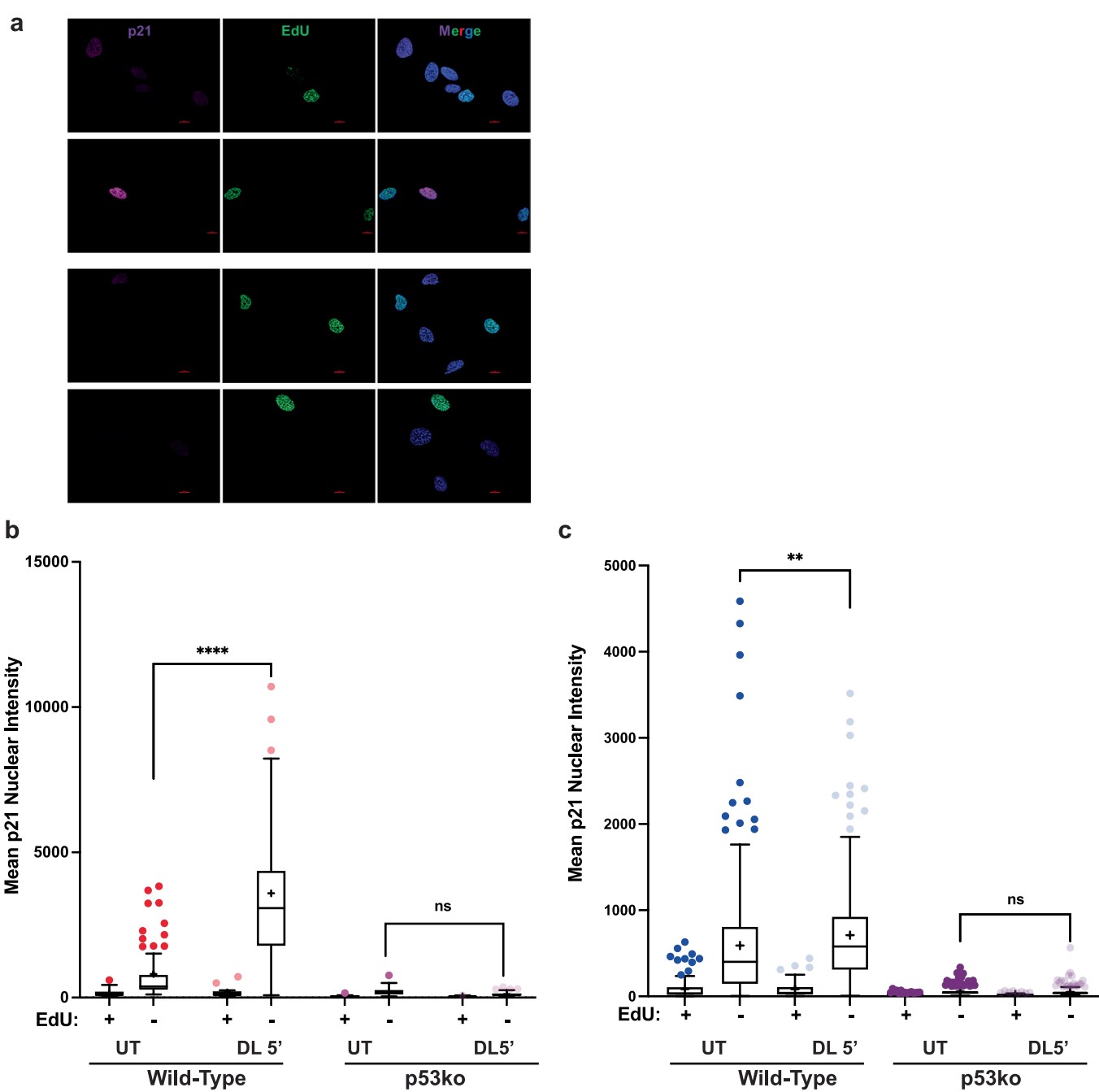

**Extended Data Fig. 6 | Telomeric 8oxoG increases p53-dependent p21 expression in non-replicating cells.** (**a-c**) 23 hours after treatment, wild-type and p53ko RPE FAP-TRF1 (**b**) and BJ FAP-TRF1 (**c**) cells were pulsed with EdU for 1 hour, and then analyzed by microscopy for p21 and EdU staining. In each condition, cells were categorized as EdU + or – populations, and the nuclear p21 signal intensity was graphed. Representative IF images are shown in panel **a**, scale bar = 10 μm. The number n of cells analyzed for each condition from two independent experiments is shown. Tukey box plot shows medians (bar), means (+), 25th to 75th interquartile range (IQR), and whiskers showing 25th or 7th percentile ± 1.5x the IQR. Data analyzed by two-way ANOVA (**p < 0.01, ****p < 0.0001).

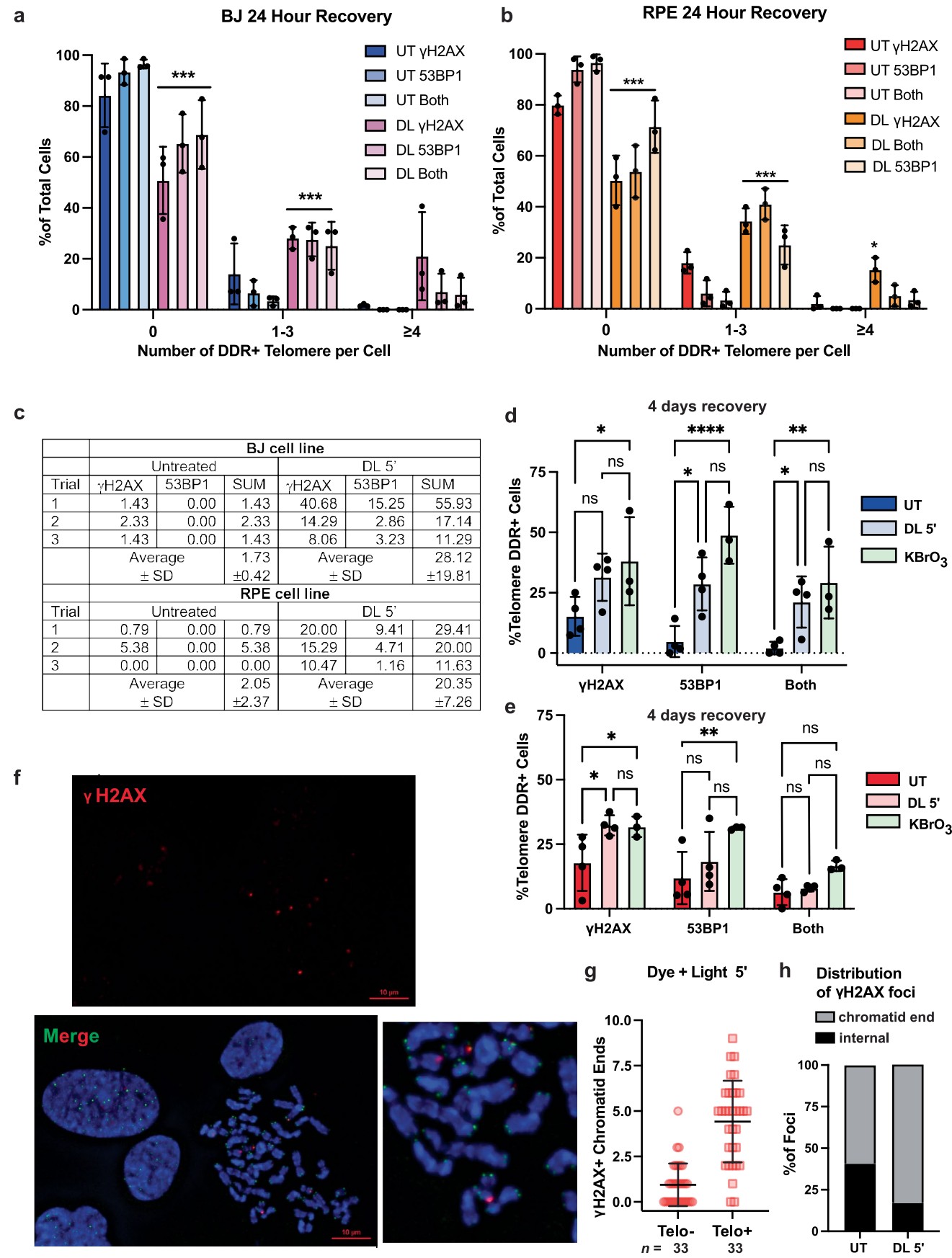

**Extended Data Fig. 7 | See next page for caption.**

**Extended Data Fig. 7 | Oxidative damage induced telomeric DDR visualized in interphase and metaphase chromosomes.** (**a-b**) Percent of cells showing 0, 1–3 or $\geq$ 4 telomeric foci co-localized with γH2AX, 53BP1 or both for BJ (**a**) and for RPE (**b**) FAP-TRF1 cells 24 hours after no treatment or 5 min DL. Error bars represent the mean $\pm$ s.d. from the number of independent experiments indicated by the black circles, in which more than 50 nuclei were analyzed per condition for each experiment. Statistical analysis was by two-way ANOVA (*p < 0.05, ***p < 0.001). (**c**) The % of cells with $\geq$ 4 γH2AX or 53BP1 positive telomeres from each experiment in Fig. 4 is shown, and summed. Data are the means and error bars are $\pm$ s.d. from three independent experiments. (**d, e**) Percent cells exhibiting telomere foci co-localized with γH2AX, 53BP1 or both for BJ (**d**) and for RPE (**e**) FAP-TRF1 cells 4 days after 5 min dye + light (DL 5') or 2.5 mM KBrO$_3$ treatment. Error bars represent the mean $\pm$ s.d. from the number of independent experiments indicated by the black circles, in which more than 50 nuclei were analyzed per condition for each experiment. Statistical analysis was by two-way ANOVA (*p < 0.05, **p < 0.01, ****p < 0.0001). (**f**) Representative image of meta-TIF chromosome spread from RPE FAP-TRF1 cell 24 hours after 5 min dye + light stained for γH2AX (red), telomere PNA (green) and DNA by DAPI (blue). Scale bar = 10 μm. (**g**) Quantification from meta-TIF assay of γH2AX positive chromatid ends lacking telomere staining (Telo -) or co-localized with telomeric PNA (Telo + ) by telo-FISH as shown in (**f**). Error bars represent the mean $\pm$ s.d. from n = 33 metaphases analyzed per condition. (**h**) Quantification from the meta-TIF assay of the distribution of γH2AX foci located at chromatid ends (telomere) versus internal (non-telomeric) sites by IF and telo-FISH as shown in panel (**f**).

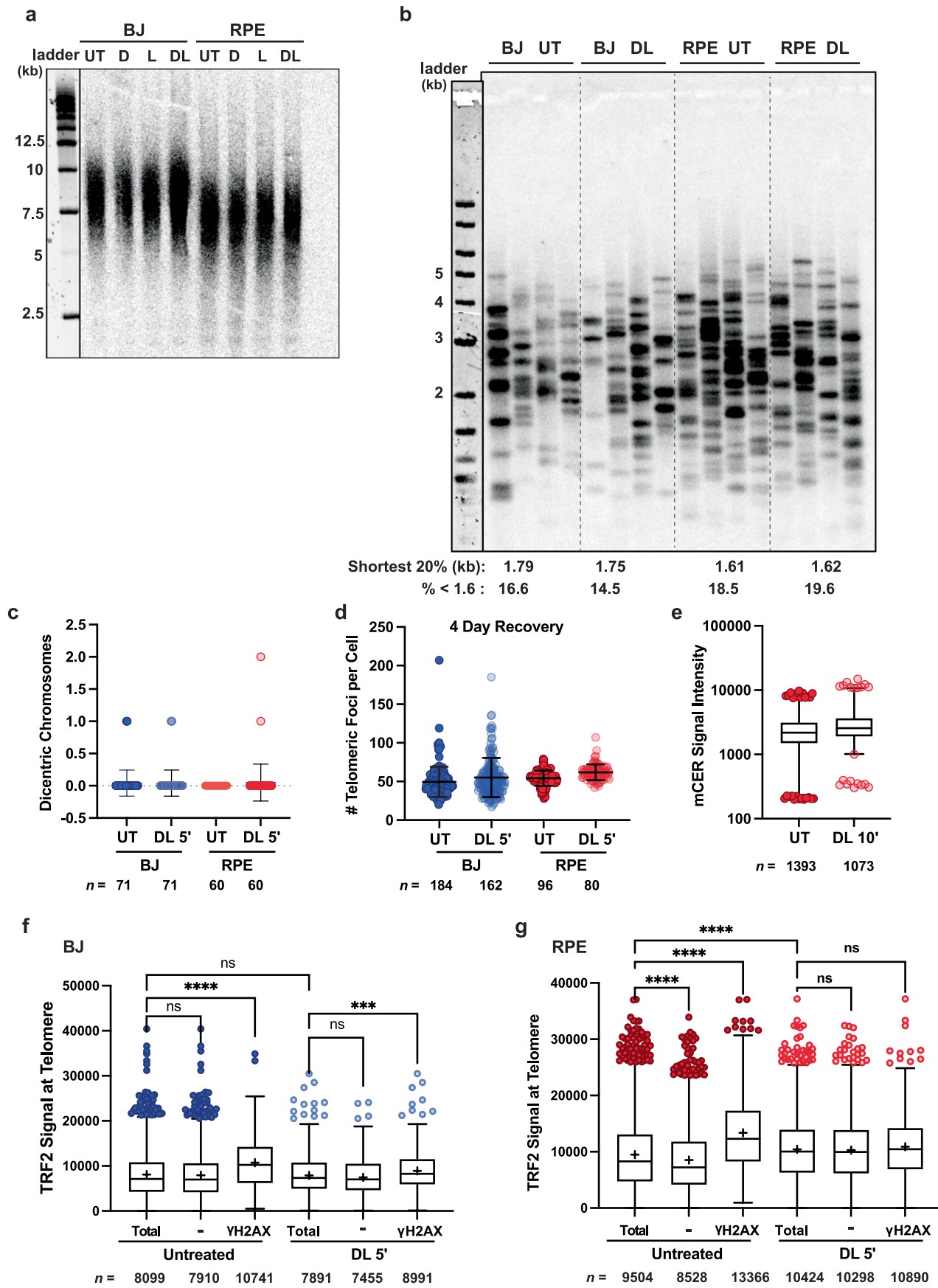

Extended Data Fig. 8 | See next page for caption.

**Extended Data Fig. 8 | Acute telomere 8oxoG damage does not cause telomere shortening or shelterin loss.** (**a**) Southern blot for telomere restriction fragment length analysis obtained from BJ and RPE FAP-TRF1 after 4 days recovery from no treatment (UT) or 5 min dye alone (D), light alone (L), or dye + light (DL) together. (**b**) TeSLA obtained from BJ and RPE FAP-TRF1 after 4 days recovery from no treatment (UT) or 5 min dye + light. Each lane is an independent PCR from the same pool of genomic DNA. Averages of telomere length for the shortest 20th percentile and the percent of telomeres shorter than 1.6 kb from two independent experiments are shown below. (**c**) Quantification of dicentric chromosomes defined as two centromeric foci for p53ko BJ (blue) and RPE (red) FAP-TRF1 cells 24 h after 5 min dye + light. (**d**) Quantification of telomere foci measured by telo-FISH for BJ (blue) and RPE (red) FAP-TRF1 cells 4 days after 5 min dye + light. For panel **c-d**, error bars represent the mean ± s.d. from the indicated *n* number of metaphases (**c**) or foci (**d**) analyzed. Statistical analysis by one-way ANOVA, all p-values were non-significant. (**e**) Quantification of mCER signal intensity per telomere foci from FAP-mCER-TRF1 in wild-type RPE FAP-TRF1 cells after no treatment (UT) or 10 min dye + light (DL 10'). Box plot shows the median and 25th to 75th interquartile range, and whiskers showing 1st to 99th percentiles. Statistical analysis by one-way ANOVA, all p-values were non-significant. (**f,g**) TRF2 colocalization with telomeres analyzed by IF and telo-FISH immediately after 5 min dye + light. DDR + telomeres were identified by γH2AX co-localization. TRF2 signal intensity was quantified per telomere foci either – or + for γH2AX in BJ (**f**) and RPE (**g**) FAP-TRF1 cells. Tukey box plot shows medians (bar), means (+), 25th to 75th interquartile range (IQR), and whiskers showing 25th or 7th percentile ± 1.5x the IQR. Statistical analysis was by Kruskal-Wallis (ns=non-significant, ***p < 0.001, ****p < 0.0001).

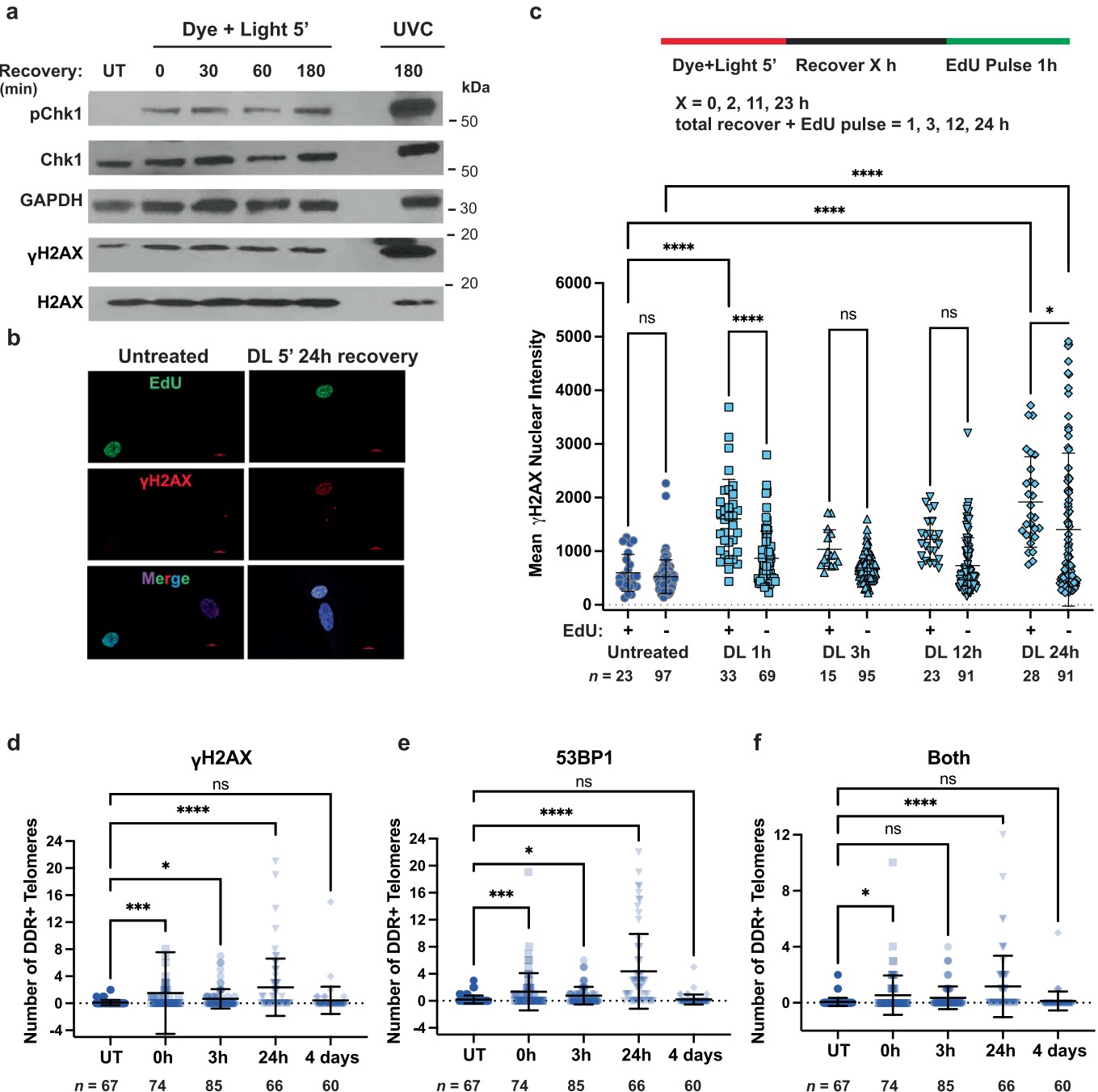

**Extended Data Fig. 9 | Time course of telomeric 8oxoG induced DDR.** (**a**) Immunoblot of phosphorylated Chk1 (S317) and H2AX (γH2AX) from untreated (UT) BJ FAP-TRF1 and cells treated with 5 min dye + light and recovered for the indicated times. UV = 20 J/m² UVC. (**b**) Representative IF image of γH2AX (red) and EdU (green) for total of 24 h recovery (23 h fresh media + 1 h EdU media). Scale bar = 10 µm. (**c**) (Top) Schematic shows experiment for EdU labeling of S-phase BJ FAP-TRF1 cells after 5 min dye + light and total recovery time. One hour before harvest after recovery time indicated by X h, cells were pulsed with EdU. (Bottom) Total nuclear γH2AX intensity as shown in (**b**) for EdU+ and EdU- cells for various total recovery time points. Error bars represent the mean ± s.d. from the indicated number n of nuclei examined. Statistical analysis by one-way ANOVA (ns=not significant, *p < 0.05, ****p < 0.0001). (**d-f**) Number of telomeres per nuclei co-localized with γH2AX, 53BP1 or both (DDR + ) in BJ FAP-TRF1 cells untreated or after 5 min dye + light and 0 h, 3 h, 24 h or 4 days recovery. Error bars represent the mean ± s.d. from the indicated number n of nuclei examined. Statistical analysis by Kruskal-Wallis (ns=not significant, *p < 0.05, ***p < 0.001, ****p < 0.0001).

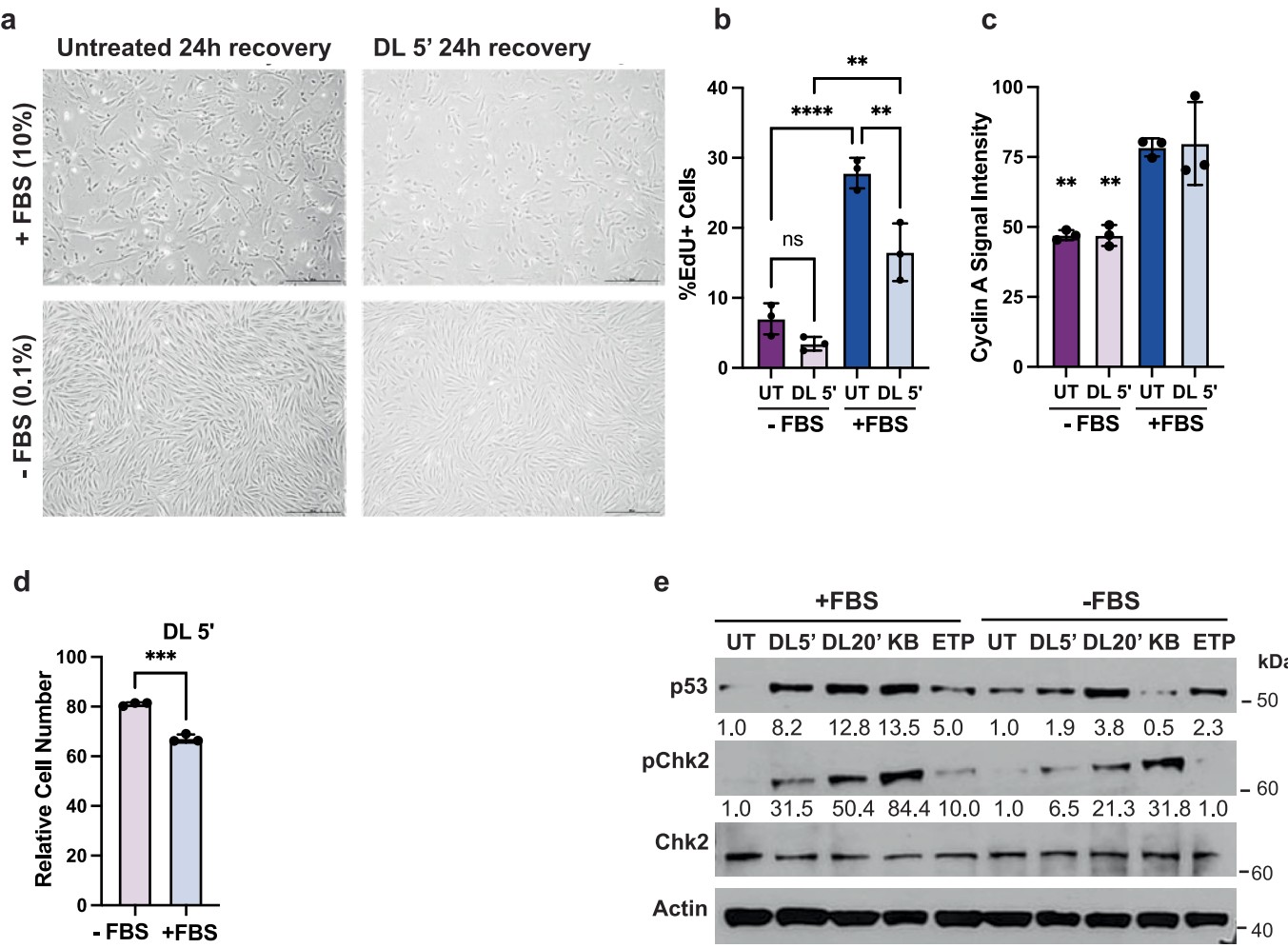

**Extended Data Fig. 10 | Suppression of replication and p53 activation in cells cultured in FBS deficient media.** (**a**) Representative brightfield images of BJ FAP-TRF1 cells grown with the indicated FBS concentration and after the indicated treatment. Scale bar = 100 μm. (**b,c**) Quantification of EdU-positive cells (**b**) and Cyclin A nuclear signal (**c**) in BJ FAP-TRF1 cells 24 hours after the indicated treatments and recovery in 10% FBS (+) or 0.1% FBS (-), as in main Fig. 6d upper panel. Error bars represent the mean ± s.d. from number of independent experiments indicated by the black circles. Statistical analysis by one-way ANOVA (**p < 0.01, ****p < 0.0001). In panel (**c**) significance is shown for -FBS cells relative to +FBS cells. (**d**) Cells were treated as described in main Fig. 6d lower panel, and counted 4 days after changing all cell culture media to 10% FBS media. Data are cell counts relative to the respective untreated control. Error bars represent the mean ± s.d. from number of independent experiments indicated by the black circles. Statistical analysis by two-tailed t-test (***p < 0.001). (**e**) Cells were cultured in 10% FBS (+) or 0.1% FBS (-) as in main Fig. 6d, upper panel, but harvested for immunoblot 3 h after treatment. KBrO₃ (KB, 2.5 mM) and etoposide (50 μM) were 1 h treatments with 3 h recovery. Numbers below p53 and pChk2 blots represent normalized protein expression.

# Reporting Summary

## Statistics

For all statistical analyses, confirm that the following items are present in the figure legend, table legend, main text, or Methods section.

| n/a | Confirmed | |
|---|---|---|
| ☐ | ☒ | The exact sample size ($n$) for each experimental group/condition, given as a discrete number and unit of measurement |
| ☐ | ☒ | A statement on whether measurements were taken from distinct samples or whether the same sample was measured repeatedly |
| ☐ | ☒ | The statistical test(s) used AND whether they are one- or two-sided<br>*Only common tests should be described solely by name; describe more complex techniques in the Methods section.* |
| ☒ | ☐ | A description of all covariates tested |
| ☒ | ☐ | A description of any assumptions or corrections, such as tests of normality and adjustment for multiple comparisons |
| ☐ | ☒ | A full description of the statistical parameters including central tendency (e.g. means) or other basic estimates (e.g. regression coefficient) AND variation (e.g. standard deviation) or associated estimates of uncertainty (e.g. confidence intervals) |
| ☐ | ☒ | For null hypothesis testing, the test statistic (e.g. $F$, $t$, $r$) with confidence intervals, effect sizes, degrees of freedom and $P$ value noted<br>*Give P values as exact values whenever suitable.* |
| ☒ | ☐ | For Bayesian analysis, information on the choice of priors and Markov chain Monte Carlo settings |
| ☒ | ☐ | For hierarchical and complex designs, identification of the appropriate level for tests and full reporting of outcomes |
| ☒ | ☐ | Estimates of effect sizes (e.g. Cohen's $d$, Pearson's $r$), indicating how they were calculated |

*Our web collection on statistics for biologists contains articles on many of the points above.*

## Software and code

Policy information about availability of computer code

| Data collection | NIS Elements AR 5.1 Nikon<br>Prism 9 Graph Pad<br>Accuri C6 Software BD<br>ImageQuant TL GE 8.2 |
|---|---|
| Data analysis | RNA sequencing data was aligned to the transcriptome with Salmon (V0.7.2) the resulting counts were analyzed with DEseq2 (V1.30.1) and enrichments calculated with FGSEA (V1.16.0), Matlab (2017A) was used for sorting genes by chromosomal coordinates and plotting |

For manuscripts utilizing custom algorithms or software that are central to the research but not yet described in published literature, software must be made available to editors and reviewers. We strongly encourage code deposition in a community repository (e.g. GitHub). See the Nature Portfolio guidelines for submitting code & software for further information.

## Data

Policy information about availability of data

All manuscripts must include a data availability statement. This statement should provide the following information, where applicable:

- Accession codes, unique identifiers, or web links for publicly available datasets
- A description of any restrictions on data availability
- For clinical datasets or third party data, please ensure that the statement adheres to our policy

All relevant data are available in the Source Data files or from the authors upon reasonable request. The mRNAseq dataset are deposited at GEO (GSE175686).

# Field-specific reporting

Please select the one below that is the best fit for your research. If you are not sure, read the appropriate sections before making your selection.

☒ Life sciences          ☐ Behavioural & social sciences          ☐ Ecological, evolutionary & environmental sciences

For a reference copy of the document with all sections, see nature.com/documents/nr-reporting-summary-flat.pdf

# Life sciences study design

All studies must disclose on these points even when the disclosure is negative.

| | |
|---|---|
| Sample size | No sample-size calculations were performed. Sample sizes were chosen based on similar published studies and standards in the field and experience. The sample size (n) is given in the figure legends. |
| Data exclusions | No data were excluded from the analyses, except for rare outliers as identified by GraphPad Prism 9 software, and indicated as blue in the Source Data. |
| Replication | Independent experiments were conducted a minimum of 3 times unless otherwise noted in the figure legends. All replications were successful. |
| Randomization | No randomization was performed. Randomization and covariates are not applicable here since we used genetically defined cell lines cultured at the same time under the same conditions, to increase robustness and control for experimental variation. |
| Blinding | Blinding was not done. Automated image analysis and quantitation was used to minimize bias. |

# Reporting for specific materials, systems and methods

We require information from authors about some types of materials, experimental systems and methods used in many studies. Here, indicate whether each material, system or method listed is relevant to your study. If you are not sure if a list item applies to your research, read the appropriate section before selecting a response.

## Materials & experimental systems

| n/a | Involved in the study |
|---|---|
| ☐ | ☒ Antibodies |
| ☐ | ☒ Eukaryotic cell lines |
| ☒ | ☐ Palaeontology and archaeology |
| ☒ | ☐ Animals and other organisms |
| ☒ | ☐ Human research participants |
| ☒ | ☐ Clinical data |
| ☒ | ☐ Dual use research of concern |

## Methods

| n/a | Involved in the study |
|---|---|
| ☒ | ☐ ChIP-seq |
| ☐ | ☒ Flow cytometry |
| ☒ | ☐ MRI-based neuroimaging |

# Antibodies

| | |
|---|---|
| Antibodies used | GFP Abcam ab6556 (1:1000 WB and IF)<br>TRF1 Abcam ab10579 (1:200 WB)<br>GAPDH Santa Cruz sc-47724 (1:30000 WB)<br>OGG1 Abcam Ab124741 (1:500)<br>Actin Cell Signaling 3700S (1:30000 WB)<br>LaminB1 Abcam ab16048 (1:500 WB and IF)<br>LaminA/C Cell Signaling 4777S (1:500 WB and IF)<br>γH2AX Santa Cruz sc-517348 (1:1000 WB, 1:250 IF)<br>53BP1 Novus NB100-304 (1:1000 IF)<br>TRF2 Novus NB110-57130 (1:500 IF)<br>MDM2 Cell Signaling 86934S (1:1000 WB)<br>p53 Santa Cruz sc-126 (1:200 WB)<br>p21 Cell Signaling 2947S (1:1000 IF and 1:2000 WB)<br>p16 Proteintech 10883-1-AP (1:200 WB)<br>pRB S807/811 Cell Signaling 8516S (1:500 WB)<br>pCHK2 T68 Cell Signaling 2197S (1:1000 WB)<br>pCHK1 S317 Cell Signaling 12302S (1:500 WB)<br>pATM S 1981 Abcam ab81292 (1:2000 WB)<br>CHK1 Cell Signaling 2360S (1:1000 WB) |

Chk2 Cell Signaling 3440 (1:1000 WB)
H3K27me3 Cell Signaling 9733 (1:500 IF)
H3K27Ac Cell Signaling  8173 (1:500 IF)
LSD1 Cell Signaling 2184 (1:500 IF)
cGAS Cell Signaling 66546 (1:200 IF)
p62 Cell Signaling 39749 (1:500 IF)

Secondary antibodies
Anti-Rabbit IgG HRP Secondary, Sigma A0545 (1:20000 WB)
Goat Anti-mouse IgG HRP antibody, Sigma A0168 (1:20000 WB)
Goat anti rabbit IgG (H+L) secondary ab Alexa fluor 488,Thermo PIA32731 (1:500 IF)
Goat anti-mouse IgG secondary Ab, Alexa Fluor 594, Thermo A32742 (1:500 IF)
Goat anti-mouse IgG secondary Ab, Alexa Fluor 647, Thermo A32728 (1:500 IF)
Goat anti-rabbit IgG secondary, Alexa Fluor 594, Thermo A32740 (1:500 IF)
Alexa Fluor 647 AffiniPure F(ab')2 Goat anti-rabbit antibody, Jackson Labs 111-606-045 (1:500 IF)

Validation | Antibodies were validated either by the manufacturer as indicated on the corresponding websites, and/or by this study in cell lines knocked out or knocked down for the gene product (OGG1, p16 and p53).

# Eukaryotic cell lines

Policy information about cell lines

Cell line source(s) |
hTERT BJ ATCC CRL-4001
hTERT RPE ATCC CRL-4000
Primary BJ ATCC CRL-2522
HeLa LT O'Sullivan lab
U2OS ATCC HTB-96
HEK29T ATCC CRL-3216

Authentication | Authentication done by ATCC for cell lines obtained from ATCC, including STR profiling.  HeLa LT cell lines was not authenticated by us for this study.

Mycoplasma contamination | All cell lines tested negative for mycoplasma contamination as confirmed by DAPI staining and microscopy, and MycoAlert elisa assay.

Commonly misidentified lines (See ICLAC register) | None

# Flow Cytometry

## Plots

Confirm that:

☒ The axis labels state the marker and fluorochrome used (e.g. CD4-FITC).

☒ The axis scales are clearly visible. Include numbers along axes only for bottom left plot of group (a 'group' is an analysis of identical markers).

☐ All plots are contour plots with outliers or pseudocolor plots.

☒ A numerical value for number of cells or percentage (with statistics) is provided.

## Methodology

Sample preparation | Cells were detached with trypsin and processed live for apoptosis measure, or fixed for cell cycle analysis.

Instrument | Accuri C6 (Beckman)

Software | Accuri C6 (Beckman)

Cell population abundance | All cells analyzed

Gating strategy | Cells gated from debris with FSC/SSC. Singlets gated with FSC-A/FSC-H.

☒ Tick this box to confirm that a figure exemplifying the gating strategy is provided in the Supplementary Information.

