## [Peer Review File · Nature Structural & Molecular Biology]

Peer Review Information

Journal: Nature Structural and Molecular Biology

Manuscript Title: Telomeric 8-Oxo-Guanine Drives Rapid Premature Senescence in the Absence of Telomere Shortening

Corresponding author name(s): Dr Patricia Opresko

Editorial Notes:

**Redactions –
unpublished data**

Reviewer Comments & Decisions:

Decision Letter, initial version:

21st Oct 2021

Dear Dr Opresko,

Thank you again for submitting your manuscript "Telomeric 8-Oxo-Guanine Drives Rapid Premature Senescence in the Absence of Telomere Shortening". I apologize for the delay while we awaited the comments (copied below) of the 3 reviewers who evaluated your paper. In light of those reports, we remain interested in your study and would like to see your response to the comments of the referees, in the form of a revised manuscript.

You will see that while Reviewer #2 is very positive about the work and suggests only minor revisions to improve data presentation, Reviewers #1 and #3 each express concerns about the physiological relevance of the findings that must be addressed by additional experimentation. Reviewer #1 requests quantitation of the level of 8oxoG formation and comparison with other oxidative lesions, if possible, as well as determination of the number of telomeres that must be damaged to trigger senescence and characterization micronuclei formation. Reviewer #3 has reservations about the level of advance obtained by the findings in the absence of additional insight into the mechanism that triggers the

senescence response to telomere damage, and whether the proposed model is relevant under physiological conditions. This reviewer also requests further characterization of the micronuclei induced by oxidative damage. Editorially, we agree that these suggestions would strengthen the work, and ask that they be incorporated in a revised manuscript. Note that we would not request that the suggested organoid models be included in the revision.

Please be sure to address/respond to all concerns of the referees in full in a point-by-point response and highlight all changes in the revised manuscript text file.

We appreciate the requested revisions are extensive. We thus expect to see your revised manuscript within 6 months. If you cannot send it within this time, please let us know. We will be happy to consider your revision as long as nothing similar has been accepted for publication at NSMB or published elsewhere. Should your manuscript be substantially delayed without notifying us in advance and your article is eventually published, the received date would be that of the revised, not the original, version.

Reporting Summary:

Please note that all key data shown in the main figures as cropped gels or blots should be presented in uncropped form, with molecular weight markers. These data can be aggregated into a single supplementary figure. While these data can be displayed in a relatively informal style, they must refer back to the relevant figures. These data should be submitted with the last revision, prior to acceptance, but you may want to start putting it together at this point.

SOURCE DATA: we urge authors to provide, in tabular form, the data underlying the graphical representations used in figures. This is to further increase transparency in data reporting, as detailed in this editorial (<http://www.nature.com/nsmb/journal/v22/n10/full/nsmb.3110.html>). Spreadsheets can be submitted in excel format. Only one (1) file per figure is permitted; thus, for multi-paneled figures,

the source data for each panel should be clearly labeled in the Excel file; alternately the data can be provided as multiple, clearly labeled sheets in an Excel file. When submitting files, the title field should indicate which figure the source data pertains to. We encourage our authors to provide source data at the revision stage, so that they are part of the peer-review process.

[REDACTED]

With kind regards,

Beth

Beth Moorefield, Ph.D.
Senior Editor
Nature Structural & Molecular Biology

Referee expertise:

Referee #1: Oxidative DNA damage/repair

Referee #2: Telomeres/senescence/DNA repair

Referee #3: Telomere maintenance/aging

Reviewers' Comments:

Reviewer #1:

Remarks to the Author:

The manuscript by Barnes et al. presents a compelling case that induction of DNA damage in the form of 8-oxoG at telomeres leads to characteristics associated with senescence. The senescence program is

associated with p53 signaling and replication stress at the telomeres. The study is thorough with rigorous interpretation of the data. The conclusions are mostly justified based on the results of the experiments. However, there is concern regarding the physiological relevance of the research as presented in this manuscript.

The technique of introducing 8oxoG at telomeres allows for characterization of cellular responses to DNA damage at specific regions of the genome, including telomeres. However, this study does not address whether 8oxoG is the predominant type of damage at telomeres in cells that are not treated with dye plus light. It would be important to describe and quantify the types of oxidized bases present at telomeres in cells and to determine if 8oxoG is the present and if it is the predominant form of telomeric damage. The guanine base can sustain DNA damage other than 8oxoG. Other types of oxidative DNA damage can arise at telomeres. The quantity of DNA damage at telomeres is also likely linked to the cellular consequences. The authors demonstrate that growth reduction is proportional to dye plus light exposure, but it is not clear how many telomeres must undergo oxidative damage for the cell to exhibit senescent features. How much telomeric damage is necessary to lead to senescent characteristics? Is damage to the telomeres of specific chromosomes important for senescence? Might the authors be able to demonstrate that cells exhibiting staining with beta-galactosidase or inducing the SASP also have oxidative damage at their telomeres? The answers to these questions are important given that many cells survive the treatment with dye plus light. Is the reason they survive because they were not damaged at their telomeres or because they responded in a manner that is not linked to senescence?

The authors perform the experiments with hTERT immortalized RPE1 cells, which are retinal epithelial cells, and BJ human foreskin fibroblasts and provide evidence consistent with the interpretation that some level of 8oxoG at the telomeres of these cells drives senescence. These cells were chosen based on their derivation from non-diseased tissue. Because senescence is associated with diseases such as cancer and also with aging, the authors should consider the use of cells that have more physiological relevance in comparison to retinal epithelial cells and human foreskin fibroblasts. Organoid models may also have greater physiological relevance.

It appears that the authors characterize cytokine levels associated with the SASP. What is not clear is whether other cytokines associated with a general inflammatory response are also present in cells harboring 8oxoG at their telomeres.

It is suggested that the authors distinguish the micronuclei they observe from the ones observed to induce genomic instability by David Pellman's group.

It is suggested that the authors perform their experiments in OGG1 knockout cells to determine if the OGG1 protein is necessary for the induction of the senescent phenotype.

Reviewer #2:

Remarks to the Author:

In this study Barnes et al capitalize on the chemoptogenetic FAP-TRF1 allele initially presented by the same team in a 2019 Molecular Cell paper. While the initial study examined how FAP-TRF1 induced oxidative damage impacts telomeres in cancer cells, the current study examines the outcome of oxidative telomere damage in a primary cell context. The cells used are primary fibroblast and RPE cells, field standards for studying telomere erosion and corresponding replicative senescence. The study relies on the ability of the team to induce specific telomere-localized oxidative damage through the expression of a FAP-TRF1 allele in the presence of a specific dye and 660 nm light. Targeted telomere oxidative damage is a major experimental leap from prior studies that used oxidative agents to induced broad non-specific damage. The study addresses if, and mechanistically how, telomere-specific oxidative damage impacts cell proliferation and/or senescence. The premise is sound, and the experiments are well thought out.

Overall, I am exceptionally impressed by the outcomes of the study. The data are robust and well controlled. I am convinced that the FAP-TRF1 allele delivers specific oxidative damage as advertised (also supported by the 2019 study). The readouts of cell proliferation, cell cycle arrest, senescence and associated phenotypes, telomere DDR activation, MIDAS, etc are all proper and congruent with expectations.

To my recollection this is the first study to induce physiologically relevant telomere-specific oxidative damage in primary cells. While prior non-directed experiments may have hinted this would result in telomere shortening, the team found instead that oxidative damage at chromosome ends induced telomere replication stress. I find this outcome more satisfying and plausible than suggestions that telomere oxidative damage was sufficient to induce rapid telomere deletions or breaks. The critical finding supports observations from multiple teams: specifically, telomere length does not regulate telomere-dependent senescence per se, but instead this regulation falls under jurisdiction of the telomere DDR. The major leap forward in this paper is demonstration that oxidative damage, through the induction of replication stress, induces telomere DDR-dependent replicative senescence independent of telomere length. This is a highly novel finding. I particularly liked the experiments in Figure 5E-G showing this requires on-going cell division.

The discussion articulates the importance of the data and was enjoyable to read. I particularly like the justified speculation related to persistent telomere DDR in cells without telomere shortening, a phenotype I have previously struggled to rationalize. The summary is appropriate and the writing is high quality throughout.

I have several minor comments below aimed to improve some aspects of the manuscript that I hope are of use to the authors.

1. Throughout the manuscript there should be a specific statement of the N for all experiments (e.g. N = 3 experiments quantifying 100 nuclei per experiment) and a specific description of the statistical measures employed. For example, Line 679: "Panel C was analyzed by unpaired T-test, and panels B, E, F, H-J by One or Two-way ANOVA." Which data were analyzed by One-way vs Two-way? This should be addressed throughout.
2. When t-tests are used, it is not always clear the data are parametric (t-tests are parametric). For example, Figure 5C-F. It may be worth revisiting to ensure all statistical tests are appropriate.
3. Throughout the manuscript there are places where statistics are not included. This includes non-significant results where a NS result is central to the premise. I suggest ensuring stats are present whenever direct comparisons are made.
4. Line 179. I do not think the stalled S-phase result is significant here, nor do I think it is germane to the argument. The key result is the overall reduction in S-phase.
5. Figure 2 Nuclear blebbing. I agree with the results. However, showing this with live imaging, if possible, would be an excellent confirmation of blebbing vs. chromosome segregation errors.
6. Lines 360 – 364 and corresponding figures. Ideally Shelterin telomere binding would be shown with ChIP, if possible, which is a more accurate readout.
7. Supp Fig 2L: Claims direct comparison between DL5' and KBrO3 but this is not shown on same axis.
8. Supp Fig 3E. I had a difficult time reading the conditions in the flow plots, it would be useful to clearly label in a figure making program.
9. Figures throughout. There are several instances where the black outline on a histogram bar is offset from the underlying bar, and/or the outline bars are of different weight/thickness in a single panel. This is distracting to some readers and the authors may want to proofread figures for cosmetic changes
10. Line 395 and 508. Cesare and Karlseder 2012 is cited here. This is a review paper. I think you may be referencing experimental data from the Cesare et al 2013 Mol Cell paper.
11. The antibodies used are not listed in the methods.

Reviewer #3:

Remarks to the Author:

In this manuscript, Barnes and colleagues report on the consequences of oxidative damage to telomeric DNA. Using an elegant method previously developed in the Opresko laboratory they applied a chemoptogenetic tool to produce an 8-oxo-guanine lesion at telomeres in fibroblasts and epithelial cells. Using this elegant system, the authors were able to define the consequences of oxidative damage at telomeres without affecting other genomic regions. The authors find that induction of this lesion at telomeric DNA is sufficient to trigger cellular senescence in a p53-dependent manner. The response was associated with activation of both ATM and ATR and markers of telomere dysfunction in proliferating cells. Interestingly, senescence was associated with micronuclei formation and was only occurring in actively dividing cells. Based on these results the authors propose a model in which oxidative stress at telomeres promotes replication-driven senescence. Notably, the senescence induced in these settings is not mediated by telomere shortening.

This manuscript is based on very well-executed work, and it's wonderfully presented. However, as it stands, the manuscript does not represent a significant leap forward in our understanding of the mechanism by which oxidative stress drives telomere defects. Previous work from the Opresko laboratory nicely showed the molecular consequences of oxidative stress at telomeres. Here, the authors apply the same experimental system to define the consequences of oxidative damage to telomeres in non-cancerous cell lines. The finding that in this context cells undergo p53-mediated senescence is interesting. However key questions remain to be addressed: What is the mechanism that triggers a p53-mediated response following oxidative damage at telomeres? Is oxidative damage at telomeres intrinsically more deleterious compared to the same damage occurring at other genomic locations? Under physiological conditions is oxidative damage to telomeres a major driver for cellular senescence?

This reviewer also has the following minor points:

-Depletion of telomere-associated proteins following oxidative stress: the authors exclude that DNA damage activation at telomeres is mediated by depletion of TRF1 or TRF2 using IF (S1 and S4). It would be more convincing to test whether the telomeres that are being detected as sites of DNA damage show a depletion of telomere-associated proteins.

-Micronuclei formation following oxidative stress: are telomeric sequences contained in the micronuclei induced by oxidative damage?

Author Rebuttal to Initial comments

We thank the reviewers for their valuable feedback. We have addressed each critique in detail and have incorporated new data and clarifications to the text, which we believe make the study stronger. We have marked major changes in red. In the revised manuscript we now followed the NSBM convention of labeling Supplementary Figures as Extended Data Figures, and now included 10 Extended Figures. We also now labeled panels with lowercase letters. Fig. 2 has been almost completely revised to include the requested experiments and data examining the mechanisms of telomeric 8oxoG induced micronuclei formation. In addition, we reorganized the manuscript and figures so that the localized telomeric DDR data (original Fig. 5, and now Fig. 4) follows the p53 DNA damage signaling data (Fig. 3). We combined the data showing telomeric 8oxoG induces telomere fragility (old Fig. 4, and now new Fig. 5) with the telomere MiDAS data (old Fig. 7, and now new Fig. 5), since both demonstrate that 8oxoG impairs telomere replication. We believe this improves the logical flow of the study, and it allowed us to include a new mechanistic model as Fig. 7 to address Reviewer #3's critique. We thank the reviewers for their time and consideration, and hope they share our enthusiasm and excitement for this revised version of the manuscript.

Reviewer #1

Remarks to Author

The manuscript by Barnes et al. presents a compelling case that induction of DNA damage in the form of 8-oxoG at telomeres leads to characteristics associated with senescence. The senescence program is associated with p53 signaling and replication stress at the telomeres. The study is thorough with rigorous interpretation of the data. The conclusions are mostly justified based on the results of the experiments. However, there is concern regarding the physiological relevance of the research as presented in this manuscript.

We thank the reviewer for the positive feedback and comments, and appreciate the recognition of the rigor with which we conducted this study. We have addressed each comment in detail.

1. The technique of introducing 8oxoG at telomeres allows for characterization of cellular responses to DNA damage at specific regions of the genome, including telomeres. However, this study does not address whether 8oxoG is the predominant type of damage at telomeres in cells that are not treated with dye plus light. It would be important to describe and quantify the types of oxidized bases present at telomeres in cells and to determine if 8oxoG is the present and if it is the predominant form of telomeric damage. The guanine base can sustain DNA damage other than 8oxoG. Other types of oxidative DNA damage can arise at telomeres.

We apologize for the misunderstanding and did not mean to imply that 8oxoG is the predominant type of damage at telomeres since this cannot be known. Although over 25 different types of oxidative DNA lesions have been described, 8oxoG is among the most common ¹. Our study does not exclude a potential role for other lesions in telomere stability, which we now acknowledge in the discussion: “These findings also raise the possibility that other oxidative lesions may contribute to telomere instability, although 8oxoG is among the most abundant” (last paragraph). However, our focus on 8oxoG as an important lesion at telomeres was due to the following:

1. We and others have shown that 8oxoG forms at telomeres under oxidative stress, not just in cells treated with our FAP-TRF1 targeting tool (dye and light) ²⁻⁹.
2. Among endogenous lesions an estimated 2,800 8oxoG form per cell per day which is two orders of magnitude greater than double strand breaks (~25), which have also been studied at telomeres ¹⁰⁻¹².
3. Mammals have evolved three distinct mechanisms to selectively process 8oxoG, highlighting the importance of this lesion in biology: 1) OGG1 removes 8oxoG paired with C, 2) MUTYH removes A mis-inserted opposite 8oxoG and 3) MTH1 removes 8oxoG from the dNTP pool to prevent insertion ^{13,14}.
4. 8oxoG's propensity to mispair with A (giving rise to G to T mutations) is attributed to ROS-associated mutational signatures 18 and 36 in the Catalogue of Somatic Mutations in Cancer (COSMIC) and contributes to carcinogenesis in mice and humans ¹⁵⁻²⁰.
5. OGG1 deficiency causes telomere dysfunction under conditions of oxidative stress ³.
6. Biochemical studies have confirmed that telomeric repeats are highly susceptible to 8oxoG formation ²¹⁻²³.

With regards to quantifying and identifying other oxidative lesions at telomeres, we have been in contact with leaders in the field of HPLC Mass-Spec methods of lesion quantification (Yinsheng Wang, Nima Mosammaparast, and Richard Wagner), and they have all informed us that you need micrograms to milligrams of DNA to measure DNA lesions. Telomeres represent less than 0.025% of the genome. From 200 micrograms of genomic DNA we are able to recover approximately 30-50 ng ²⁴, which is woefully insufficient for such studies. Unfortunately, we are unaware of any technology that currently exists to identify and quantify the distribution of oxidative lesions at telomeres. Furthermore, we are unaware of any reports that have successfully quantified the oxidative hydantoin lesions in the genome, presumably because they are rare.

2. The quantity of DNA damage at telomeres is also likely linked to the cellular consequences. The authors demonstrate that growth reduction is proportional to dye plus light exposure, but it is not clear how many telomeres must undergo oxidative damage for the cell to exhibit senescent features. How much telomeric damage is necessary to lead to senescent characteristics?

We agree with the reviewer that the amount of damage induced correlates with the amount of senescence. We show a greater induction of senescence with 20 min of dye and light exposure compared to 5 minutes exposure (**Fig. 1h-i**). We demonstrated in a methods manuscript that the length of dye and light exposure is proportional to the amount of telomeric 8oxoG produced in the telomeres (Fig. 3B, Barnes, et al.,

Targeted Formation of 8-oxoguanine in Telomeres. *Methods Molecular Biology*, Vol. 2444, Nima Mosammaparast (Eds): DNA Damage Responses, Chapter 9, accepted). In this assay 8oxoG in the telomeres are detected by converting 8oxoG in telomere restriction fragments to a double strand break with FPG glycosylated and S1 nuclease *in vitro*². Cleaved telomeres are detected as faster migrating species by gel electrophoresis, and the amount cleavage correlates to lesion frequency. We now conducted this assay in BJ-hTERT and RPE-hTERT cells after dye and light exposure and show 20 minutes of dye and light indeed produces more telomere 8oxoG than 5 minutes, which is above background (new **Extended Data Fig. 1f-g**). Importantly, 40 mM KBrO₃ treatment resulted in telomere 8oxoG levels comparable to dye and light treatment (new **Extended Data Fig. 1g**) whereas 2.5 and 10 mM were below the level of detection for this assay (data not shown). Due to lack of sensitivity (see comment #1) we are unable to precisely quantify the exact number of 8oxoG per telomere but continue to pursue this.

Our results indicate that in cells it is the conversion of 8oxoG to fragile and dysfunctional telomeres that triggers cellular senescence, since non-replication cells are largely unaffected by telomeric 8oxoG induction (**Fig. 6**). Given the multiple repair and lesion bypass mechanisms a cell can employ; we do not expect every 8oxoG lesion will be converted into a dysfunctional telomere. However, our data is consistent with previous reports that 4-5 dysfunctional telomeres (detected as DDR foci at telomeres) are sufficient to trigger cell senescence²⁵.

3. Is damage to the telomeres of specific chromosomes important for senescence? Might the authors be able to demonstrate that cells exhibiting staining with beta-galactosidase or inducing the SASP also have oxidative damage at their telomeres? The answers to these questions are important given that many cells survive the treatment with dye plus light. Is the reason they survive because they were not damaged at their telomeres or because they responded in a manner that is not linked to senescence?

Given the stochastic nature of 8oxoG formation, we do not expect that some telomeres would be more susceptible than others. Whether specific telomeres are more susceptible to replication stress is a fascinating question but beyond the scope of our manuscript and is likely influenced by cell type. Given the lag time in the appearance of senescent phenotypes (24 hr to 4 days) as is common, the initial 8oxoG lesion is expected to be processed, repaired or diluted through replication during the course of the experiments. We predict that cells that did not undergo telomeric 8oxoG-induced senescence were able to successfully process the 8oxoG (rapid repair or replication bypass) to prevent replication stress and fragility. However, our data indicate that those cells in which 8oxoG led to telomere dysfunction, as indicated by 53BP1 DDR foci at telomeres (**Fig. 4**), went on to become senescent. To confirm this, we expected cells showing 53BP1 foci after dye and light exposure would also be positive for SA-beta-galactosidase. For this we tried two fluorescent beta-gal substrates, but these reagents did not work for immuno-fluorescence microscopy in our hands. Instead, we used p53 immuno-fluorescence staining as a surrogate marker for senescent cells, since p53 is required for telomeric 8oxoG-induced senescence (**Fig.**

3g). We now show that 3 hrs after 5 min of telomeric 8oxoG formation, p53 positive cells show a greater induction of 53BP1 foci compared to p53 negative cells (new **Extended Data Fig. 5l,m**). We used Nutlin, which activates p53 by preventing interaction with MDM2, as a positive control for increased p53 staining. However, Nutlin did not induce 53BP1 foci, confirming that 53BP1 DDR was specific for telomere damage and was not a general response to p53 activation.

4. The authors perform the experiments with hTERT immortalized RPE1 cells, which are retinal epithelial cells, and BJ human foreskin fibroblasts and provide evidence consistent with the interpretation that some level of 8oxoG at the telomeres of these cells drives senescence. These cells were chosen based on their derivation from non-diseased tissue. Because senescence is associated with diseases such as cancer and also with aging, the authors should consider the use of cells that have more physiological relevance in comparison to retinal epithelial cells and human foreskin fibroblasts. Organoid models may also have greater physiological relevance.

We favor the view that aging is not a “disease” per se, but rather evidence indicates that the accumulation of senescent cells contributes to diseases that occur with aging. Senescence has been shown to arise in non-diseased tissues and cells in animal models. Indeed, many cancers bypass telomere driven senescence through p53 inactivation to achieve immortalization. There is a wealth of literature that indicates human skin fibroblasts and retinal epithelial cells are susceptible to senescence²⁶, and both cell types are expected to experience singlet oxygen through UVA generated processes. An organoid model is an interesting idea to pursue in the future. Based on the exciting results we obtained with the BJ and RPE cells, we secured pilot funding to generate a transgenic mouse model that allows us to express the FAP-TRF1 in any tissue, which will allow us to examine roles for telomeric 8oxoG in promoting cellular senescence in vivo and in tissue. But these experiments will require at least 2-3 years to optimize and complete and are beyond the scope of this manuscript.

5. It appears that the authors characterize cytokine levels associated with the SASP. What is not clear is whether other cytokines associated with a general inflammatory response are also present in cells harboring 8oxoG at their telomeres.

The purpose of examining SASP was to strengthen the conclusion that telomeric 8oxoG induces senescence, since SASP is a hallmark of senescence. We agree that roles for telomeric 8oxoG in immune signaling is an exciting topic to pursue, but we are unclear how analysis of other cytokines in this study would help to further support or strengthen our conclusions regarding the induction of senescence. However, we do have a separate project underway in the lab examining roles for telomeric 8oxoG and telomere dysfunction in immune signaling.

6. It is suggested that the authors distinguish the micronuclei they observe from the ones observed to induce genomic instability by David Pellman’s group.

Excellent idea. This is particularly exciting because the manuscript we believe the reviewer is referring to²⁷, examined the role of breakage-fusion-bridge cycles from dicentric chromosomes in driving chromosomal instability in p53 deficient RPE-hTERT cells. In our study we primarily examine the RPE-

hTERT cells that are p53 proficient, but we see no evidence for chromatid bridges, or dicentric chromosomes in the p53ko metaphase spread experiments. Therefore, we propose the primary mechanism of micronuclei formation may be blebbing of chromatin to generate cytoplasmic chromatin foci as described by others²⁸. To confirm this, we now performed live cell imaging and show dye and light treatment does not increase mitoses which produce micronuclei (new **Fig. 2i,h** and **Supplemental Movies**). Staining fixed cells for centromere and telomere DNA also showed dye and light treatment increased acentric fragments, but not lagging chromosomes, which are associated with mitotic errors (new **Fig. 2g**). The fixed cells also revealed no damage-induced increase in chromatid bridge (new **Fig. 2f**). Moreover, consistent with our fixed cell data, live cell imaging revealed micronuclei which blebbed from the primary nucleus in interphase cells following dye and light treatment. While this does not exclude other possibilities, we believe the micronuclei we observe are consistent with mechanisms of CCF generation associated with senescent cells.

7. It is suggested that the authors perform their experiments in OGG1 knockout cells to determine if the OGG1 protein is necessary for the induction of the senescent phenotype.

Excellent suggestion. [REDACTED]

Reviewer #2

Remarks to Author

In this study Barnes et al capitalize on the chemoptogenetic FAP-TRF1 allele initially presented by the same team in a 2019 Molecular Cell paper. While the initial study examined how FAP-TRF1 induced oxidative damage impacts telomeres in cancer cells, the current study examines the outcome of oxidative telomere damage in a primary cell context. The cells used are primary fibroblast and RPE cells, field standards for studying telomere erosion and corresponding replicative senescence. The study relies on the ability of the team to induce specific telomere-localized oxidative damage through the expression of a FAP-TRF1 allele in the presence of a specific dye and 660 nm light. Targeted telomere oxidative damage is a major experimental leap from prior studies that used oxidative agents to induced broad non-specific damage. The study addresses if, and mechanistically how, telomere-specific oxidative damage impacts cell proliferation and/or senescence. The premise is sound, and the experiments are well thought out.

Overall, I am exceptionally impressed by the outcomes of the study. The data are robust and well controlled. I am convinced that the FAP-TRF1 allele delivers specific oxidative damage as advertised (also supported by the 2019 study). The readouts of cell proliferation, cell cycle arrest, senescence and associated phenotypes, telomere DDR activation, MIDAS, etc are all proper and congruent with expectations.

To my recollection this is the first study to induce physiologically relevant telomere-specific oxidative

damage in primary cells. While prior non-directed experiments may have hinted this would result in telomere shortening, the team found instead that oxidative damage at chromosome ends induced telomere replication stress. I find this outcome more satisfying and plausible than suggestions that telomere oxidative damage was sufficient to induce rapid telomere deletions or breaks. The critical finding supports observations from multiple teams: specifically, telomere length does not regulate telomere-dependent senescence per se, but instead this regulation falls under jurisdiction of the telomere DDR. The major leap forward in this paper is demonstration that oxidative damage, through the induction of replication stress, induces telomere DDR-dependent replicative senescence independent of telomere length. This is a highly novel finding. I particularly liked the experiments in Figure 5E-G showing this requires on-going cell division.

The discussion articulates the importance of the data and was enjoyable to read. I particularly like the justified speculation related to persistent telomere DDR in cells without telomere shortening, a phenotype I have previously struggled to rationalize. The summary is appropriate and the writing is high quality throughout.

We thank the reviewer for the positive feedback, and greatly appreciate the shared excitement regarding our discovery of a novel mechanism of rapid oxidative telomere damage induced senescence. We agree that our data could help explain persistent telomeric DDR that has been observed in vivo in the absence of telomere shortening.

I have several minor comments below aimed to improve some aspects of the manuscript that I hope are of use to the authors.

1. Throughout the manuscript there should be a specific statement of the N for all experiments (e.g. N = 3 experiments quantifying 100 nuclei per experiment) and a specific description of the statistical measures employed. For example, Line 679: "Panel C was analyzed by unpaired T-test, and panels B, E, F, H-J by One or Two-way ANOVA." Which data were analyzed by One-way vs Two-way? This should be addressed throughout.

We have carefully edited all the figure legends to make sure N is included for all the experiments. We also clearly indicate which statistics were conducted for each panel and experiment. Additions are marked in red.

2. When t-tests are used, it is not always clear the data are parametric (t-tests are parametric). For example, Figure 5C-F. It may be worth revisiting to ensure all statistical tests are appropriate. Thank you for the comments. We re-analyzed the quantification of telomere losses and fragile telomeres by Mann-Whitney since the data is not parametric, which was confirmed in Graphpad.

3. Throughout the manuscript there are places where statistics are not included. This includes non-significant results where a NS result is central to the premise. I suggest ensuring stats are present whenever direct comparisons are made.

We ensure that the statistical test was stated for each experiment. To conserve space we summarized statistics at the end of the figure legend in some cases. We included “ns” where applicable.

4. Line 179. I do not think the stalled S-phase result is significant here, nor do I think it is germane to the argument. The key result is the overall reduction in S-phase.

We also did not observe a change the fraction of G1 and G2 cells after dye and light. Therefore, for completeness we thought it important to include the stalled S-phase fraction, in which we also did not see a change.

5. Figure 2 Nuclear blebbing. I agree with the results. However, showing this with live imaging, if possible, would be an excellent confirmation of blebbing vs. chromosome segregation errors. Excellent suggestion. We have now conducted live cell imaging with cells expressing H2B-RFP and conducted staining for centromeric and telomeric DNA in micronuclei of fixed cells (please see new **Fig. 2f-g** and **Supplemental movies**, and response to Reviewer 1 comment #6).

6. Lines 360 – 364 and corresponding figures. Ideally Shelterin telomere binding would be shown with ChIP, if possible, which is a more accurate readout.

We agree that ChIP can be useful readout for Shelterin binding. But, given that only a few telomeres (4-5) showed DDR foci after telomeric 8oxoG damage, we thought it was important to examine individual telomeres, rather than telomeres in bulk. Also, based on the evidence that 5 min of dye and light induces approximately 1-5 8oxoG per telomeres², we predicted this would be insufficient to displace the bulk of shelterin at the telomeres. Additionally, we added new data to examine TRF2 signal intensity by IF at DDR+ and DDR- telomeres after 8oxoG damage (new **Extended Data Fig. 8f**). The DDR+ telomeres do not show a reduction in TRF2.

7. Supp Fig 2L: Claims direct comparison between DL5' and KBrO3 but this is not shown on same axis. We believe the reviewer was referring to the comparison that we made between the reduction in relative cell number induced by DL5' (Fig. 1C) and 2.5 and 10 mM KBrO3 exposure (old Supp Fig 1L, now Extended Data Fig. 2g), since there was no Supp Fig 2L. To clarify we added “2.5 mM KBrO3 treatment for one hour reduced BJ and RPE FAP-TRF1 cell growth to levels comparable with five minutes dye and light treatment (compare **Figs. 1c-d** with **Extended Data Fig. g,h**)”. While the data are not on the same axis, the different treatments are normalized to untreated cells. We wished to point out that the reduction in cell number induced by DL5' (to 53% BJ and 72% RPE) was more similar to that achieved with 2.5 mM KBrO3 (to 52% BJ AND 82% RPE), rather than with 10 mM KBrO3 or etoposide which were more toxic.

8. Supp Fig 3E. I had a difficult time reading the conditions in the flow plots, it would be useful to clearly label in a figure making program.

We believe the reviewer is referring to the flow plots in old Supplementary Fig 2E. We enlarged this plot and moved this data to new **Extended Data Fig. 2j**, and enlarged the labels.

9. Figures throughout. There are several instances where the black outline on a histogram bar is offset from the underlying bar, and/or the outline bars are of different weight/thickness in a single panel. This is distracting to some readers and the authors may want to proofread figures for cosmetic changes

We have rebuilt the figures using Adobe Illustrator and edited the figures for consistency regarding font size and weight of the lines in graphs.

10. Line 395 and 508. Cesare and Karlseder 2012 is cited here. This is a review paper. I think you may be referencing experimental data from the Cesare et al 2013 Mol Cell paper.

Yes, the reviewer is correct. Thank you for noting this.

11. The antibodies used are not listed in the methods.

We have now added the list of antibodies to the methods.

Reviewer #3:

Remarks to the Author:

In this manuscript, Barnes and colleagues report on the consequences of oxidative damage to telomeric DNA. Using an elegant method previously developed in the Opresko laboratory they applied a chemoptogenetic tool to produce an 8-oxo-guanine lesion at telomeres in fibroblasts and epithelial cells. Using this elegant system, the authors were able to define the consequences of oxidative damage at telomeres without affecting other genomic regions. The authors find that induction of this lesion at telomeric DNA is sufficient to trigger cellular senescence in a p53-dependent manner. The response was associated with activation of both ATM and ATR and markers of telomere dysfunction in proliferating cells. Interestingly, senescence was associated with micronuclei formation and was only occurring in actively dividing cells. Based on these results the authors propose a model in which oxidative stress at telomeres promotes replication-driven senescence. Notably, the senescence induced in these settings is not mediated by telomere shortening.

This manuscript is based on very well-executed work, and it's wonderfully presented. However, as it stands, the manuscript does not represent a significant leap forward in our understanding of the mechanism by which oxidative stress drives telomere defects. Previous work from the Opresko laboratory nicely showed the molecular consequences of oxidative stress at telomeres. Here, the authors apply the

same experimental system to define the consequences of oxidative damage to telomeres in non-cancerous cell lines. The finding that in this context cells undergo p53-mediated senescence is interesting.

We greatly appreciate the reviewer's positive comments that our work was "very well-executed" and "wonderfully presented". We hope with our additional experiments and clarification that we were able to convince the reviewer of the significant advance our work represents, and why we are so excited about our discoveries.

1. However key questions remain to be addressed: What is the mechanism that triggers a p53-mediated response following oxidative damage at telomeres?

We demonstrated that the mechanism of p53 and DDR activation at telomere is not by shelterin loss (new **Extended Data Fig. 8f,g**), telomeres shortening (**Extended Data Fig. 8a,b**), or telomere loss (**Fig. 5a-f**), but rather by induction of telomere fragility (**Fig. 5a-f**). Previous work has shown that telomere fragility induced by TRF1 loss can trigger senescence in mouse cells²⁹. Further, we show activation of DDR at telomeres by 8oxoG induction depends on replication, since the dye and light treatment did not induce a significant increase in DDR foci or in SA-beta-gal positive cells when damage was induced in quiescent (G0), non-replicating cells (**Figs. 6f-g**). The induction of mitotic DNA synthesis is further evidence that telomeric 8oxoG causes replication stress at telomeres (**Fig. 5g,h**). We now include a model to clarify the mechanism, which underscores the difference in how cancerous and non-cancerous cells respond to oxidative damage at telomeres (see **Fig. 7**).

2. Is oxidative damage at telomeres intrinsically more deleterious compared to the same damage occurring at other genomic locations?

It is very difficult to conduct a direct (apples to apples) comparison between telomeres and another genomic region, because telomeres as a group represent 0.025% of the genome, and while very low, they are more abundant than a single copy gene or specific gene promoter. Perhaps G-quadruplex forming sequences or common fragile sites could be examined as a group, but we would need a way to target damage only to these sites. We believe development and validation of such a tool is beyond the scope of this manuscript, but is a very exciting future direction.

However, we agree with the wealth of data from the literature that show telomeres are particularly difficult regions of the genome to replicate, and therefore, resemble common fragile site sequences in that they are more sensitive to replication stress (most recently review in³⁰). We find that telomeres are sensitive to replication stress induced by 8oxoG formation, and that similar to replication stress and telomere fragility induced by TRF1 loss, this is sufficient to induce a DDR at telomeres and trigger rapid senescence.

To address this further, we now compared targeted telomeric 8oxoG production to KBrO_3 treatment, since although KBrO_3 is a general oxidant, it primarily induces 8oxoG in the genome. Using a dose which resulted in similar senescence to 5 min dye and light, we found very similar inductions of telomere DDR (new data added to **Fig 4**). This suggests that even though telomeres are a small fraction of the genome (and thus a small target), they are highly susceptible to general oxidative stress, consistent with previous reports (see above, and response to reviewer 1 comment #1). Furthermore, we show some telomeric DDR foci persist 4 days after treatment (new **Extended Data Fig. 7d,e**). By comparing KBrO_3 treatment (general oxidant) to target telomeric damage with the FAP tool, our data support the proposal that oxidative damage to telomeres is “intrinsically more deleterious” than other “general” genomic regions, since damaging only telomeres is sufficient to produce senescence.

3. Under physiological conditions is oxidative damage to telomeres a major driver for cellular senescence?

This is an excellent question. Numerous studies in the literature indicate that there are multiple mechanisms of senescence induction under physiological conditions (reviewed in ^{31,32}). We propose the 8oxoG formation is likely linked to an oxidative stress induced form of cellular senescence. As pointed out by reviewer #2, our data support observations from multiple teams regarding persistent DDR positive telomeres in senescent cells caused by oxidative stress, in the absence of shortening (see ³³⁻³⁸). Our data indicate that these DDR positive telomeres are not just “collateral damage” but provide direct evidence that oxidatively damaged telomeres drive senescence. We believe our data provide a reasonable mechanism by which oxidative damage a telomere triggers senescence in the absence of shortening.

This reviewer also has the following minor points:

1. Depletion of telomere-associated proteins following oxidative stress: the authors exclude that DNA damage activation at telomeres is mediated by depletion of TRF1 or TRF2 using IF (S1 and S4). It would be more convincing to test whether the telomeres that are being detected as sites of DNA damage show a depletion of telomere-associated proteins.

Thank you for this suggestion. We added new data showing that telomeres exhibiting a DDR (γH2AX staining) do not show a significant reduction in TRF2 signal intensity by IF (new **Extended Data Fig. 8f, g**).

-Micronuclei formation following oxidative stress: are telomeric sequences contained in the micronuclei induced by oxidative damage?

Yes, we have now quantified the fraction of MN showing telomeric DNA after telomeric 8oxoG induction (**Fig. 2g**). More than 75% of the MN stain positive for telomeric DNA.

- 1 Evans, M. D., Dizdaroglu, M. & Cooke, M. S. Oxidative DNA damage and disease: induction, repair and significance. *Mutat Res* **567**, 1-61, doi:10.1016/j.mrrev.2003.11.001 (2004).
- 2 Fouquerel, E. *et al.* Targeted and Persistent 8-Oxoguanine Base Damage at Telomeres Promotes Telomere Loss and Crisis. *Mol Cell* **75**, 117-130 e116, doi:10.1016/j.molcel.2019.04.024 (2019).
- 3 Wang, Z. *et al.* Characterization of oxidative guanine damage and repair in mammalian telomeres. *PLoS Genet* **6**, e1000951, doi:10.1371/journal.pgen.1000951 (2010).
- 4 Baquero, J. M. *et al.* Small molecule inhibitor of OGG1 blocks oxidative DNA damage repair at telomeres and potentiates methotrexate anticancer effects. *Sci Rep* **11**, 3490, doi:10.1038/s41598-021-82917-7 (2021).
- 5 Wu, J., McKeague, M. & Sturla, S. J. Nucleotide-Resolution Genome-Wide Mapping of Oxidative DNA Damage by Click-Code-Seq. *J Am Chem Soc* **140**, 9783-9787, doi:10.1021/jacs.8b03715 (2018).
- 6 Poetsch, A. R. The genomics of oxidative DNA damage, repair, and resulting mutagenesis. *Comput Struct Biotechnol J* **18**, 207-219, doi:10.1016/j.csbj.2019.12.013 (2020).
- 7 O'Callaghan, N., Baack, N., Sharif, R. & Fenech, M. A qPCR-based assay to quantify oxidized guanine and other FPG-sensitive base lesions within telomeric DNA. *Biotechniques* **51**, 403-411, doi:10.2144/000113788 (2011).
- 8 An, N., Fleming, A. M., White, H. S. & Burrows, C. J. Nanopore detection of 8-oxoguanine in the human telomere repeat sequence. *ACS Nano* **9**, 4296-4307, doi:10.1021/acsnano.5b00722 (2015).
- 9 Barnes, R. P., Fouquerel, E. & Opresko, P. L. The impact of oxidative DNA damage and stress on telomere homeostasis. *Mech Ageing Dev* **177**, 37-45, doi:10.1016/j.mad.2018.03.013 (2019).
- 10 Tubbs, A. & Nussenzweig, A. Endogenous DNA Damage as a Source of Genomic Instability in Cancer. *Cell* **168**, 644-656, doi:10.1016/j.cell.2017.01.002 (2017).
- 11 Dilley, R. L. *et al.* Break-induced telomere synthesis underlies alternative telomere maintenance. *Nature* **539**, 54-58, doi:10.1038/nature20099 (2016).
- 12 Doksani, Y. & de Lange, T. Telomere-Internal Double-Strand Breaks Are Repaired by Homologous Recombination and PARP1/Lig3-Dependent End-Joining. *Cell Rep* **17**, 1646-1656, doi:10.1016/j.celrep.2016.10.008 (2016).
- 13 De Rosa, M., Johnson, S. A. & Opresko, P. L. Roles for the 8-Oxoguanine DNA Repair System in Protecting Telomeres From Oxidative Stress. *Frontiers in Cell and Developmental Biology* **9**, doi:10.3389/fcell.2021.758402 (2021).
- 14 Markkanen, E. Not breathing is not an option: How to deal with oxidative DNA damage. *DNA Repair (Amst)* **59**, 82-105, doi:10.1016/j.dnarep.2017.09.007 (2017).
- 15 Alexandrov, L. B. *et al.* The repertoire of mutational signatures in human cancer. *Nature* **578**, 94-101, doi:10.1038/s41586-020-1943-3 (2020).
- 16 Viel, A. *et al.* A Specific Mutational Signature Associated with DNA 8-Oxoguanine Persistence in MUTYH-defective Colorectal Cancer. *EBioMedicine* **20**, 39-49, doi:10.1016/j.ebiom.2017.04.022 (2017).
- 17 van den Boogaard, M. L. *et al.* Defects in 8-oxo-guanine repair pathway cause high frequency of C > A substitutions in neuroblastoma. *Proc Natl Acad Sci U S A* **118**, doi:10.1073/pnas.2007898118 (2021).
- 18 Brady, S. W. *et al.* Pan-neuroblastoma analysis reveals age- and signature-associated driver alterations. *Nat Commun* **11**, 5183, doi:10.1038/s41467-020-18987-4 (2020).

- 19 Sakumi, K. *et al.* Ogg1 knockout-associated lung tumorigenesis and its suppression by Mth1 gene disruption. *Cancer Res* **63**, 902-905 (2003).
- 20 Nakabeppu, Y. Cellular levels of 8-oxoguanine in either DNA or the nucleotide pool play pivotal roles in carcinogenesis and survival of cancer cells. *Int J Mol Sci* **15**, 12543-12557, doi:10.3390/ijms150712543 (2014).
- 21 Oikawa, S. & Kawanishi, S. Site-specific DNA damage at GGG sequence by oxidative stress may accelerate telomere shortening. *FEBS Lett* **453**, 365-368 (1999).
- 22 Oikawa, S., Tada-Oikawa, S. & Kawanishi, S. Site-specific DNA damage at the GGG sequence by UVA involves acceleration of telomere shortening. *Biochemistry* **40**, 4763-4768, doi:bi002721g [pii] (2001).
- 23 Henle, E. S. *et al.* Sequence-specific DNA cleavage by Fe²⁺-mediated fenton reactions has possible biological implications. *J Biol Chem* **274**, 962-971 (1999).
- 24 Parikh, D., Fouquerel, E., Murphy, C. T., Wang, H. & Opresko, P. L. Telomeres are partly shielded from ultraviolet-induced damage and proficient for nucleotide excision repair of photoproducts. *Nat Commun* **6**, 8214, doi:10.1038/ncomms9214 (2015).
- 25 Kaul, Z., Cesare, A. J., Huschtscha, L. I., Neumann, A. A. & Reddel, R. R. Five dysfunctional telomeres predict onset of senescence in human cells. *EMBO Rep* **13**, 52-59, doi:embor2011227 [pii] 10.1038/embor.2011.227 (2012).
- 26 Coppe, J. P., Desprez, P. Y., Krtolica, A. & Campisi, J. The senescence-associated secretory phenotype: the dark side of tumor suppression. *Annu Rev Pathol* **5**, 99-118, doi:10.1146/annurev-pathol-121808-102144 (2010).
- 27 Umbreit, N. T. *et al.* Mechanisms generating cancer genome complexity from a single cell division error. *Science* **368**, doi:10.1126/science.aba0712 (2020).
- 28 Ivanov, A. *et al.* Lysosome-mediated processing of chromatin in senescence. *J Cell Biol* **202**, 129-143, doi:10.1083/jcb.201212110 (2013).
- 29 Sfeir, A. *et al.* Mammalian telomeres resemble fragile sites and require TRF1 for efficient replication. *Cell* **138**, 90-103, doi:S0092-8674(09)00721-1 [pii] 10.1016/j.cell.2009.06.021 (2009).
- 30 Glousker, G. & Lingner, J. Challenging endings: How telomeres prevent fragility. *Bioessays* **43**, e2100157, doi:10.1002/bies.202100157 (2021).
- 31 Campisi, J., Andersen, J. K., Kapahi, P. & Melov, S. Cellular senescence: a link between cancer and age-related degenerative disease? *Semin Cancer Biol* **21**, 354-359, doi:10.1016/j.semcancer.2011.09.001 (2011).
- 32 Hernandez-Segura, A., Nehme, J. & Demaria, M. Hallmarks of Cellular Senescence. *Trends Cell Biol* **28**, 436-453, doi:10.1016/j.tcb.2018.02.001 (2018).
- 33 Anderson, R. *et al.* Length-independent telomere damage drives post-mitotic cardiomyocyte senescence. *EMBO J* **38**, doi:10.15252/embj.2018100492 (2019).
- 34 Fumagalli, M. *et al.* Telomeric DNA damage is irreparable and causes persistent DNA-damage-response activation. *Nat Cell Biol*, doi:ncb2466 [pii] 10.1038/ncb2466 (2012).
- 35 Jurk, D. *et al.* Chronic inflammation induces telomere dysfunction and accelerates ageing in mice. *Nat Commun* **2**, 4172, doi:10.1038/ncomms5172 (2014).
- 36 Lagnado, A. *et al.* Neutrophils induce paracrine telomere dysfunction and senescence in ROS-dependent manner. *EMBO J*, e106048, doi:10.15252/embj.2020106048 (2021).

- 37 Michaloglou, C. *et al.* BRAFE600-associated senescence-like cell cycle arrest of human naevi. *Nature* **436**, 720-724, doi:nature03890 [pii] 10.1038/nature03890 (2005).
- 38 Victorelli, S. *et al.* Senescent human melanocytes drive skin ageing via paracrine telomere dysfunction. *EMBO J* **38**, e101982, doi:10.15252/embj.2019101982 (2019).

Decision Letter, first revision:

10th Mar 2022

Dear Patty,

Thank you for submitting your revised manuscript "Telomeric 8-Oxo-Guanine Drives Rapid Premature Senescence in the Absence of Telomere Shortening" (NSMB-A45283A). It has now been seen by the original referees and their comments are below. The reviewers find that that the revisions fully address their prior concerns, and therefore we'll be happy in principle to publish it in Nature Structural & Molecular Biology, pending minor revisions to incorporate the Referee #2's minor requests and to comply with our editorial and formatting guidelines.

Please don't hesitate to contact me if you have any questions.

With kind regards,

Beth

Beth Moorefield, Ph.D.
Senior Editor
Nature Structural & Molecular Biology

Reviewer #1 (Remarks to the Author):

The authors have adequately addressed the comments of this Reviewer. This manuscript will be of interest to a wide range of scientists.

Reviewer #2 (Remarks to the Author):

Thank you to the authors for addressing the reviewers' comments. I am satisfied with the responses and suggest publication of the manuscript. Congratulations to the team on a successful study.

There were a series of minor issues the authors may want to address before publication.

Figure 2f - there are no lines representing the mean for the UT RPE FAP and DL FAP data points.

Figures 2g, Extended Data Figure 5i, and Extended data figure 10d - Relative cell number is not set to 100% in the control / Untreated samples as it is elsewhere in the manuscript.

Figure 5c-f – the y axis is labelled “number per metaphase”. This should result in the data grouping in in quanta of 1, 2, 3, etc as observed in figure 5i. This is not the case for the data presented. Is there a mistake in the axis label?

Extended data figure 5m - The data points are missing from the 5th column on the right graph

Extended data figure 10b, c = in the legend states “In panel (b) significance is shown for – FBS relative to + FBS cells”. Do the authors mean (c) not (b)?

Reviewer #3 (Remarks to the Author):

The authors addressed all my concerns in full.

Author Rebuttal, first revision:

We thank reviewer #2 for the additional feedback and have addressed each comment as follows:

1. Figure 2f - there are no lines representing the mean for the UT RPE FAP and DL FAP data points.

Thanks for catching this. We fixed the error.

2. Figures 2g, Extended Data Figure 5i, and Extended data figure 10d - Relative cell number is not set to 100% in the control / Untreated samples as it is elsewhere in the manuscript.

Figure 2g shows the proportion of MN positive for centromere or telomere DNA. The numbers are the ratio of the positive MN divided by the total number of MN. That is why the UT is not 100%.

In extended figure 5i and 10d, we are showing the percent of cell growth of the treated cells, and simply did not show that the UT was set to 100%. We modified the figure legends to make this more clear.

3. Figure 5c-f – the y axis is labelled “number per metaphase”. This should result in the data grouping in in quanta of 1, 2, 3, etc as observed in figure 5i. This is not the case for the data presented. Is there a mistake in the axis label?

For the metaphase spread experiments, we normalized all number to 46 chromosomes. This results in numbers with decimal values. We clarified this in the methods.

4. Extended data figure 5m - The data points are missing from the 5th column on the right graph

The nutlin treated RPE cells responded so strongly, that none classified as p53 negative. This is why there are no data points.

5. Extended data figure 10b, c = in the legend states “In panel (b) significance is shown for – FBS relative to + FBS cells”. Do the authors mean (c) not (b)?

Yes. Thank you for catching this mistake.

Final Decision Letter:

16th May 2022

Dear Dr. Opresko,

We are now happy to accept your revised paper "Telomeric 8-Oxo-Guanine Drives Rapid Premature Senescence in the Absence of Telomere Shortening" for publication as a Article in Nature Structural & Molecular Biology.

Over the next few weeks, your paper will be copyedited to ensure that it conforms to Nature Structural & Molecular Biology style. Once your paper is typeset, you will receive an email with a link to choose the appropriate publishing options for your paper and our Author Services team will be in touch regarding

any additional information that may be required.

As soon as your article is published, you can generate your shareable link by entering the DOI of your article here: http://authors.springernature.com/share.

Corresponding authors will also receive an automated email with the shareable link

Note the policy of the journal on data deposition:

<http://www.nature.com/authors/policies/availability.html>.

Your paper will be published online soon after we receive proof corrections and will appear in print in the next available issue. You can find out your date of online publication by contacting the production team shortly after sending your proof corrections. Content is published online weekly on Mondays and Thursdays, and the embargo is set at 16:00 London time (GMT)/11:00 am US Eastern time (EST) on the day of publication. Now is the time to inform your Public Relations or Press Office about your paper, as they might be interested in promoting its publication. This will allow them time to prepare an accurate and satisfactory press release. Include your manuscript tracking number (NSMB-A45283B) and our journal name, which they will need when they contact our press office.

About one week before your paper is published online, we shall be distributing a press release to news organizations worldwide, which may very well include details of your work. We are happy for your institution or funding agency to prepare its own press release, but it must mention the embargo date and Nature Structural & Molecular Biology. If you or your Press Office have any enquiries in the

meantime, please contact press@nature.com.

Please note that *Nature Structural & Molecular Biology* is a Transformative Journal (TJ). Authors may publish their research with us through the traditional subscription access route or make their paper immediately open access through payment of an article-processing charge (APC). Authors will not be required to make a final decision about access to their article until it has been accepted. [Find out more about Transformative Journals](https://www.springernature.com/gp/open-research/transformative-journals)

Authors may need to take specific actions to achieve [compliance](https://www.springernature.com/gp/open-research/funding/policy-compliance-faqs) with funder and institutional open access mandates. If your research is supported by a funder that requires immediate open access (e.g. according to [Plan S principles](https://www.springernature.com/gp/open-research/plan-s-compliance)) then you should select the gold OA route, and we will direct you to the compliant route where possible. For authors selecting the subscription publication route, the journal's standard licensing terms will need to be accepted, including [self-archiving policies](https://www.springernature.com/gp/open-research/policies/journal-policies). Those licensing terms will supersede any other terms that the

author or any third party may assert apply to any version of the manuscript.

Sincerely,

Carolina Perdigoto, PhD
Chief Editor
Nature Structural & Molecular Biology
orcid.org/0000-0002-5783-7106